# A global, spherical, finite-element model for postseismic deformation using *Abaqus*

Grace A. Nield[1,2], Matt A. King[1], Rebekka Steffen[3], Bas Blank[4]

[1]Surveying and Spatial Sciences, School of Technology, Environments and Design, University of Tasmania, Australia
[2]Department of Geography, Durham University, Durham, UK.
[3]Lantmäteriet, Gävle, Sweden
[4]Faculty of Aerospace Engineering, Delft University of Technology, Delft, NL

*Correspondence to*: Grace A. Nield (grace.a.nield@durham.ac.uk)

**Abstract.** We present a finite-element model of postseismic solid Earth deformation built in the software package *Abaqus*
(version 2018). The model is global and spherical, includes self-gravitation and is built for the purpose of calculating
postseismic deformation in the far-field (>~300km) of major earthquakes. An earthquake is simulated by prescribing slip on a
fault plane in the mesh and the model relaxes under the resulting change in stress. Both linear Maxwell and biviscous (Burgers)
rheological models have been implemented and the model can be easily adapted to include different rheological models and
lateral variations in Earth structure, a particular advantage over existing models. We benchmark the model against an analytical
coseismic solution and an existing open-source postseismic model code, demonstrating good agreement for all fault geometries
tested. Due to the inclusion of self-gravity the model has the potential for predicting deformation in response to multiple
sources of stress change, for example, changing ice thickness in tectonically active regions.

## 1 Introduction

Earthquakes cause deformation at the Earth's surface from the immediate coseismic fault slip and, thereafter, from several
processes. Afterslip on the fault on or near the rupture zone (e.g. Huang et al., 2014) and poroelastic relaxation due to changes
in fluid pressure (e.g. Masterlark and Wang, 2002) result in deformation of near-field locations (typically <300km from the
fault). As the Earth responds to stress changes associated with the earthquake, the lower crust and upper mantle undergo
prolonged postseismic viscoelastic deformation. This is the dominant mode of displacement in the far-field and, for very large
earthquakes (> magnitude 9), this can be observed geodetically over 1000 km from the earthquake source (Shao et al., 2016)
and be sustained over decades. For example, deformation from the 1960 magnitude 9.5 Chile earthquake was still being
observed ~40 years after the event (Khazaradze and Klotz, 2003). Postseismic deformation can be observed by increasingly
dense networks of geodetic measurements such as Global Positioning System (GPS) (e.g. Freed et al., 2012) and
Interferometric Synthetic Aperture Radar (InSAR) (e.g. Wang and Fialko, 2018) motivating the development of models to
help interpret geodetically-observed deformation. The study of postseismic deformation can provide useful information about
the Earth and can be used to place constraints on the inferred Earth structure (Pollitz, 2005) or rheology (Freed et al., 2012). It

is important to be able to accurately quantify postseismic deformation in regions where deformation occurs as a result of multiple sources, for example in Alaska where the Earth is also deforming in response to a changing ice load (Suito and Freymueller, 2009).

Postseismic deformation has been studied using a variety of modelling techniques, from simple layered half-space (also known as "flat-Earth") models to models that consider Earth's sphericity and three-dimensional material properties. For studying the far-field effects of earthquakes consideration of sphericity is required (Pollitz, 1992). Pollitz (1997) developed a software (*VISCO1D*, freely available from the USGS at: https://www.usgs.gov/software/visco1d, which contains a link to download the software) to calculate postseismic deformation on a spherical layered Earth incorporating an Earth model that varies in the

radial direction only. It uses viscoelastic normal mode theory and calculates deformation from spherical harmonic expansion of global modes of relaxation. Deformation can be calculated both with and without the effect of gravity, that is, the restoring forces within the Earth allowing it to regain gravitational equilibrium. The effect of including gravity is that deformation ceases once the Earth has adjusted to the change in stress, whereas without gravity, deformation continues.  This model has been employed for a wide range of earthquake studies, such as constraining Earth properties and rheological behaviour in response

to the magnitude 7.9 2002 Denali earthquake in Alaska  (Pollitz, 2005), computing postseismic gravity change following the magnitude 9.1 2011 Tohoko-Oki earthquake gravity change (Han et al., 2014), and modelling postseismic deformation of Antarctica (King and Santamaría-Gómez, 2016). Furthermore, *VISCO1D* has been used to benchmark other codes of postseismic deformation calculation (Wang et al., 2006). Whilst *VISCO1D* is a powerful tool, it is limited to a one-dimensional Earth model which may not be appropriate for some studies where lateral variations in Earth structure may be required to fully

explain geodetic observations (King and Santamaría-Gómez, 2016).

Several finite-element models of postseismic deformation have also been developed (Freed et al., 2006; Masterlark et al., 2001; Takeuchi and Fialko, 2013), although the majority have been restricted to a half-space geometry. Hu and Wang (2012) updated their finite-element model (Hu et al., 2004) to include spherical geometry and used it to study the 2004 magnitude 9.2 Sumatra

earthquake, where a spherical geometry allowed for more realistic slab and fault geometry to be included, however they limited the spatial extent of the model to the case study region. Agata et al. (2019) modelled postseismic deformation following the 2011 magnitude 9.0 Tohoku-Oki earthquake with a finite-element model incorporating spherical (but not global) geometry, restricting the model domain to 2500 km by 2500 km to allow for a high-resolution mesh. Whilst limiting the spatial extent is computationally efficient and does not cause a problem for the estimation of postseismic deformation, to compute a fully self-

gravitational solution using spherical harmonics a full global model is required. Self-gravitation takes account of the change in gravity field caused by deformation and the redistribution of mass within the Earth. The effect of self-gravitation due to earthquake-related deformation is likely to be small for the majority of earthquakes, and not on the same scale as that caused by glacial isostatic adjustment since the magnitude of deformation is much smaller, but including it allows consistent modelling of deformation due to multiple sources (e.g. glacial cycles). Furthermore, for large earthquakes, the change in gravity field

resulting from deformation and redistribution of mass within the Earth could be significant, and, if the earthquake occurs under the ocean, redistribution of water also contributes to the change in gravity and hence affects sea levels (Broerse et al., 2011).

The purpose of the model presented in this paper is to estimate far-field postseismic deformation primarily from large earthquakes. The model is a global, spherical finite-element model constructed with the commercial software *Abaqus*

(https://www.3ds.com/products-services/simulia/products/abaqus/; Hibbitt et al., 2016). It is an improvement on existing models since it includes both global spherical geometry and the capability to use 3D variations in Earth properties which can significantly affect viscoelastic deformation (e.g. Latychev et al., 2005; Zhong et al., 2003, Wu et al., 2013). Furthermore, the model has the potential to simultaneously include different sources of Earth deformation, for example postseismic deformation in a region undergoing, or which has previously undergone, surface-mass change (e.g., ice-mass change). This is needed, for

instance, to separate present-day surface deformation signals observed by GPS.

This paper describes the model setup and the methods implemented to estimate coseismic and postseismic deformation (Sects. 2 and 3). Since afterslip and poroelastic relaxation are considered only to affect deformation in the near-field of the fault for all but the largest earthquakes (Peña et al., 2019) they are not included in our model. We benchmark model results for several

simple scenarios against existing codes (Sect. 4) and discuss the advantages and limitations of the model (Sect. 5). The input files are available to allow other users to replicate results and use the methods to set up their own postseismic deformation models.

## 2 Model Setup

### 2.1 Model Geometry and Mesh

The model is a global, spherical representation of the Earth, developed in *Abaqus* version 2018, but is also compatible with older versions of the software as well. It is based on the finite-element model for computing glacial isostatic adjustment following the methods described by Wu (2004), whereby we use the same set up of a viscoelastic Earth relaxing in response to a stress change, but instead of applying a changing ice-surface load we implement a fault and associated slip. We model a

spherical Earth rather than taking a layered half-space approach so a more realistic geometry can be included and gravity perturbations due to deformation can be accounted for by means of a spherical harmonic expansion. The model has layers from the surface to the core-mantle boundary with computational feasibility being the only limit on the number of layers. Each layer is assigned material properties according to user requirements and would typically consist of an elastic outer layer representing the uppermost lithosphere, with viscoelastic lower lithosphere and mantle layers below.


An earthquake is simulated in the model by movement on a fault plane within the mesh (see Sect. 3 for details) and in order to compute deformation to a sufficient accuracy a high-resolution mesh is required in the vicinity of the fault. To balance the need for a high-resolution mesh against a model with a computationally feasible number of elements, we take the approach of constructing two separate parts (*Abaqus* keyword *PART): one for the high-resolution fault region and one for the lower resolution surrounding earth (Fig. 1). The two parts are then tied together (*Abaqus* keyword *Tie) using surface-to-surface tie constraints. This means that although the two separate meshes have non-conforming elements relative to each other, the tie constraints ensure there is no relative movement between the surfaces and that displacement and stress are continuous through the boundaries. Tie coefficients are generated and used to interpolate quantities from nodes on one side of the mesh to nodes on the other side of the mesh. The size of the part containing the fault and high resolution mesh can be changed according to the requirements of the individual case study, and in the simulations presented in this study, is large enough to contain the far-field area of interest. The high-resolution mesh block extends to 670 km depth (i.e. the base of the upper mantle) so that any deformation that may be caused by the fault movement is within the block and hence there is minimal stress to transfer across the tied surfaces. The element type used is an 8-node linear brick element (C3D8 in *Abaqus*).

## 2.2 Rheology and Earth Parameters

All layers within the model are assigned a density (ρ), Poisson's ratio (ν) and Young's Modulus (E) (*Abaqus* key words *Density *Elastic). For viscoelastic layers below the purely elastic outer shell, a relaxation time is specified which is based on a Prony time series (*Abaqus* key words *Viscoelastic). In this study we limit our benchmarking examples to a 1D linear viscoelastic rheology with one (Maxwell) or two (Burgers) relaxation times and include one example implementing the elastic properties of the widely used Preliminary Reference Earth Model (PREM, Dziewonski & Anderson, 1981). However, *Abaqus* has the capability of implementing a variety of rheological models, including user-specified constitutive equations. For example, Freed et al. (2012) combined power-law rheology with a transient phase to model postseismic deformation following the 1999 magnitude 7.1 Hector Mine earthquake, and van der Wal et al. (2010) used a composite rheology based on laboratory-derived flow laws for diffusion and dislocation creep (Hirth and Kohlstedt, 2003; Karato and Wu, 1993) to model global glacial isostatic adjustment. Furthermore, variations of our model could be constructed using a 3D Earth structure (e.g. van der Wal et al., 2015) (more information on how this can be done is included in the supplementary material (Nield, 2022)).

## 2.3 Boundary Conditions

We follow the approach described in Section 4.1 of Wu (2004) and apply elastic foundations (*Abaqus* keyword *Foundation) to each layer boundary with a material density contrast occurring across it (including the surface and core-mantle-boundary). This means that advection of pre-stress is included and takes care of the restoring forces of buoyancy neglected in a conventional finite-element model (Wu, 2004, equation 3). The elastic foundations have a stiffness equal to the difference in density multiplied by gravitational acceleration (see Wu (2004) equations 12a,b,c). Schmidt et al. (2012) show that the use of

foundations at surfaces not perpendicular to the direction of gravity produces errors in results, but we ensure that density contrasts occur only at spherical layer boundaries in our model which satisfies this requirement. The foundations could be also replaced by spring elements when an inclined density contrast is used (Schmidt et al., 2012).

## 2.4 Time Steps

Once the model has been set up with the geometry, rheology, Earth parameters, boundary conditions, and fault slip (see Sect. 3); time steps are created to run the model. The first step of the run is a static step to allow the fault to slip. In this step all material properties are treated as elastic and the fault displaces immediately, hence no actual time needs to be assigned. All subsequent steps are for the simulation of postseismic relaxation and consequently must have a time assigned to them. One full run of all the time steps is one iteration.

## 2.5 Self-Gravitation

Movement of mass due to deformation of the Earth perturbs the gravity field which in turn affects the Earth's deformation. The effect of changes in the gravity field needs to be taken into account to make the model self-gravitating. This is done iteratively as described in Section 4.3 of Wu (2004), first using a non-self-gravitating model and computing the resulting gravitational potential from the radial displacement using the equations presented in Section 4 of Wu (2004). Applying this as a new load at the interfaces of the model where a density contrast occurs across them, the displacement is recalculated (i.e. another iteration is run). Wu (2004, Section 4.3) suggests running the model for 4-5 iterations to achieve convergence of the solution.

## 2.6 Outputs

*Abaqus* can compute many different output variables depending on the model setup. For our model the main output of interest is a global grid of deformation in the east, north and radial directions. It is also possible to output stress, strain, and perturbation to the geoid which could be used to correct satellite gravimetry data for the gravity change associated with very large earthquakes (e.g. Han et al. 2014).

## 3 Implementation of an Earthquake

### 3.1 Fault Geometry and Slip

In our model an earthquake is simulated by prescribing slip on a fault plane. In order to implement this in a finite-element mesh we require two separate surfaces that can move relative to each other. We take the approach of Steffen et al. (2014), whose model simulates fault slip due to changes in stress during glacial cycles, whereby the fault plane geometry is created prior to meshing and then once generated, the mesh is subsequently altered to produce two surfaces. This is accomplished by

duplicating the nodes that lie on the fault surface and then reassigning the duplicated nodes to the elements on one side of the fault plane, thereby creating a "cut" in the mesh. Although the elements on each side of the fault are defined by different nodes, the node pairs initially have the same coordinates (Fig. 2).

The model of Steffen et al. (2014) incorporates a complex stress history comprising tectonic stresses and stresses relating to glacial cycles allowing faults to slip in response to the changing conditions. Our model differs from this approach because the amount of slip that occurs on the fault plane is prescribed. This negates the need to specify any fault surface parameters such as coefficient of friction or cohesion but does require detailed knowledge of the earthquake event. Prescribing an amount of slip on the fault plane is a reasonable approach as fault properties, such as length, width, strike, dip, rake, and slip are often

solved through independent means such as fault inversion studies (Hayes, 2017) and detailed fault slip information is not required when studying far-field deformation. Furthermore, we do not include any pre-stress in the model, that is, there is no tectonic or background stress applied before the fault slips. This has no effect for models that use Maxwell or Burgers rheology as they are independent of stress.

## 3.2 Coseismic Slip

Once the finite-element mesh has been adjusted to accommodate the fault plane, coseismic displacement is prescribed following the approach of Masterlark (2003). For every node pair on the fault plane (i.e. nodes on either side of the fault that have the same coordinates), a third "dummy" node is created at the same location but not connected to any element in the mesh (Fig. 2). Dummy nodes are assigned a displacement boundary condition (*Abaqus* keyword *Boundary) in the x and y directions of a local coordinate system aligned with the fault plane (see Fig. 2) with the amount of displacement in each direction

depending on the slip and rake. For example, using basic trigonometry, a slip of 5m at a rake of -60° would equate to slip of 2.5m in the X direction and 4.3m in the Y direction. Kinematic constraint equations (*Abaqus* keyword *Equation) are then constructed for each node pair and its corresponding dummy node which specifies the relative displacement between the node pair in the X and Y direction. There is no relative displacement in the Z direction of the local coordinate system (i.e. normal to the fault plane), as the fault is not allowed to open. Because the constraint equations are specified separately for each node,

the fault plane can accommodate a spatially variable slip distribution. The first step of the model run is a static step (*Abaqus* keyword *Static) in which the nodes (and hence the fault plane) displace according to the kinematic constraint equations. During the static step, all materials are treated as elastic.

## 3.3 Postseismic Deformation

Following the first step of the model run (i.e. the earthquake), the subsequent time steps simulate postseismic deformation

(*Abaqus* keyword *Visco) by allowing the mesh to deform under the stresses caused by the displacement on the fault plane. The kinematic constraint equations remain in place throughout the model run which means that there is no further relative displacement between the node pairs.

## 4 Benchmarking

In order to verify the model we benchmark the results against those produced with the Okada analytical solution for coseismic displacement (Okada, 1985) and the *VISCO1D* code (Version 3) for gravitational postseismic relaxation (Pollitz, 1997).

### 4.1 Test Setup

We perform three benchmarking tests each with different geometry: a strike-slip fault, a 30° dipping reverse fault, and a 45° dipping fault with rake of -60°, as summarised in Table 1. All faults outcrop at the surface of the model. The resolution of the mesh on the fault plane is $10 \times 5$ km. In plan view, the *Abaqus* mesh has 10 km elements in the vicinity of the fault, increasing

to 50 km at the edge of the fault region, with elements in the surrounding low-resolution part, starting 1000 km away from the fault, being up to 500 km (Fig. 1). The element size increases with depth as shown in Table 2. For the 30° dipping reverse fault we tested a coarser and a finer mesh and found that our chosen mesh resolution provided a satisfactory trade-off between computation time and accuracy, with the finer mesh giving only small differences in near- and far-field displacement compared to our chosen mesh, for a 10-fold increase in computation time. The results of this sensitivity test are discussed in more detail

in Appendix A.

The model is run for 500 years to verify both the short and long term postseismic deformation, and for 4 iterations to ensure convergence of the self-gravitational solution. The *Abaqus* input files for the benchmarking tests are included in the supplementary material (Nield, 2022). *VISCO1D* is run for the same time period and with maximum spherical harmonic degree

of 2000, equivalent to ~10 km resolution.

In all benchmarking tests the same simple Earth structure is used (details for the *Abaqus* model are in Table 2), which is based on the Earth structure used by Pollitz (1997) for the upper most 670 km, and we include a uniform lower mantle layer below. We found that the vertical deformation results for *VISCO1D* were sensitive to the number of layers in the input Earth structure

and the presence of a lower mantle, so whilst the material properties are the same for both models, the *VISCO1D* Earth model contains more layers. We use this simple Earth structure to ensure our results are consistent with the Okada analytical solution for a homogenous half-space and the results presented by Pollitz (1997). We use a linear Maxwell rheology for all three benchmarking tests. We additionally verify the implementation of Burgers rheology against *VISCO1D* for the third test, with the transient viscosity (η) given in Table 2. Furthermore, we run the third test case in *Abaqus* using the elastic properties from

PREM, keeping the same three-layer Maxwell viscosity profile as the benchmarking Earth model (Table 2). We do not benchmark the results from this test case as the Okada analytical solution is only valid for a homogeneous Earth. The input files for each test case and the Earth models used in the *VISCO1D* modelling for both Maxwell and Burgers rheology are given in the supplementary material (Nield, 2022). All output displacements are shown normalised to the fault slip, in other words as a percentage of fault slip. This is consistent with the presentation of results by Pollitz (1997) and provides a useful metric

for comparison of results. Differences between the model results are given as the difference between the percentages of fault slip.

## 4.2 Coseismic Results

The results of the benchmarking exercises are shown in Figs. 3-5. We show coseismic displacement in the east, north and vertical directions for a profile perpendicular to the fault strike for each of the benchmarking tests. Surface displacement is
shown in Appendix B, Figs. B1-B3. The Okada analytical solution is shown by the dark green line with the equivalent *Abaqus* coseismic output shown by the light green dots. The difference between the models is shown in panel b) of Figs. 3-5. Overall, there is an excellent agreement between the *Abaqus* model and the Okada analytical solution. Of the three cases, the strike-slip fault agrees most closely (Fig. 3), with differences peaking at 1% of the fault slip in the north direction (Fig. 3b). There are larger differences visible for the dipping fault cases (Figs. 4-5). Some of the near-field finer details of displacement in the
east and vertical directions for the dipping faults (i.e. the directions with most displacement) are not perfectly captured by *Abaqus*, for example at 50 km from the fault (see Fig. 4a, vertical direction), and we attribute this to the mesh resolution and the distorted element shape that is required to mesh around a dipping fault. However, these differences are a maximum of 6% (Fig. 4b) and at distances greater than 300 km from the fault this decreases to less than 0.5%. Since the aim of our model is to predict far-field deformation these differences are acceptable.

## 4.3 Postseismic Results

Figs. 3-5 also show profiles of the postseismic displacement in the east, north and vertical directions for *VISCO1D* (light blue line) and the *Abaqus* model (dark blue dashed line). We show displacement as a percentage of fault slip at three times: 10, 50 and 100 years after the earthquake with corresponding differences between the models shown in panel b) of each figure. Surface postseismic displacement at each of these times is shown in Appendix B, Figs. B1-B3. The *Abaqus* predictions closely
agree with *VISCO1D* for the strike-slip fault case (Fig. 3). Small differences can be observed for the dipping faults particularly in the vertical direction (Figs. 4b and 5b). The mismatch in the coseismic displacement due to limitations in mesh resolution is the cause of mismatch for the postseismic displacement, i.e., less coseismic displacement from the *Abaqus* model would result in less stress and therefore less relaxation. Slight improvements in the near-field displacement could be made by increasing the mesh resolution in the vicinity of the fault but would come at a computational cost (see Appendix A). However,
the differences remain small and are concentrated within 100 km of the fault with a maximum difference of 5% of the fault slip after 100 years of relaxation (Fig. 4b) and less than 0.5% at distances greater than 300 km from the fault. Therefore, we conclude that the model results are reliable for far-field postseismic deformation.

Figs. 6-7 show displacement through time for the fault geometry in test 3 for two points at 100 km (Fig. 6) and 300 km (Fig.
7) from the fault using both a Maxwell and Burgers rheology. There is clear consistency in the evolution of postseismic relaxation between our model and *VISCO1D*, for both rheologies (compare solid with dashed lines on Figs. 6a and 7a). The

insets in Figs. 6-7 show the results for the initial 30 years of the model run where a rapid displacement can be observed from the transient viscosity of the Burgers rheology (orange/red lines) before converging with the results from the Maxwell model (blue lines) after approximately 30 years. Over the 500-year period, differences between our *Abaqus* model and *VISCO1D* for

the Burgers rheology are less than 0.6% of the fault slip within 100 km of the fault (Fig. 6b), or less than 0.1% at distances of 300 km or more (Fig. 7b). For the Maxwell rheology, differences peak at 0.5% of the fault slip within 100 km, reducing to 0.2% at 300 km.

Figs. 6a and 7a also show the displacement for test case 3 with the PREM elastic properties (green dotted lines). At 100 km

distance from the fault (Fig, 6a), the different elastic structure results in slightly more coseismic displacement than the simple elastic structure used in the benchmarking tests, which results in greater postseismic displacement with time. At 300 km distance from the fault (Fig. 7a), the depth varying PREM elastic structure results in only very small differences. The surface coseismic and postseismic displacements at 10, 50 and 100 years after the earthquake are shown in Fig. B4, Appendix B.

## 5 Discussion and Conclusions

We have presented a finite-element model constructed in *Abaqus* for the purposes of modelling far-field coseismic and postseismic deformation. The model is global, spherical and self-gravitating, and allows for simple modification to include three-dimensional Earth structure, non-linear rheologies and alternative or multiple sources of stress change.

The model performs well when compared with the Okada coseismic analytical solution and predictions from the postseismic

*VISCO1D* program for all three fault scenarios we have tested. For the coseismic displacement, differences are less than 6% of the fault slip, with the largest differences in the vertical direction and near to the fault; in the fault far-field the differences are negligible. For the postseismic displacement, differences are less than 5% of the fault slip and at distances of 300 km from the fault, i.e. the far-field which is the focus of our model, this reduces to differences of 0.5%. Furthermore, we have verified that the evolution of displacement through time is an excellent match with *VISCO1D* for both Maxwell and Burgers rheologies.

For interest, we also computed the displacement for test case 3 using the PREM elastic properties, finding differences in results within 100 km of the fault. Although, in the far-field, changing the elastic structure has minimal impact on the resulting postseismic displacement as we used the same viscosity structure.

Inclusion of self-gravitation makes only a small difference to the displacement, peaking at 0.2 mm (less than 0.1% of the fault

slip) for the reverse fault (test 2) in the vertical displacement and is negligible in the horizontal directions. This demonstrates that it is not necessary to include self-gravitation when modelling postseismic deformation in isolation, but it could become important when modelling postseismic deformation alongside other larger sources of deformation such as changes in ice loading.

The main limitation of the model at present is that the geometry is restricted to a single fault plane within the mesh and it cannot have multiple segments of a fault plane with different strike or dip. This is due to the difficulties of constructing a valid spherical mesh in *Abaqus* with brick elements that conform to verification checks. For the same reason, including more than one earthquake in the same model would be problematic. This issue could potentially be resolved by employing external meshing software (e.g., Gmsh (Geuzaine and Remacle, 2009)). In the case of a fault inversion that suggests multiple fault

segments (e.g. Ye et al., 2014), an approximation of all the fault planes into a single geometry could still provide a realistic far-field estimate of postseismic deformation (e.g. Takeuchi and Fialko, 2013), particularly if the fault geometry and slip are adjusted so that model output matches observations of coseismic displacement (e.g. Sun et al., 2018). Far-field postseismic deformation is less sensitive to simplifications made to the fault geometry and slip distribution than near-field deformation (Khazaradze et al., 2002, Tregoning et al., 2013, Zhou et al. 2012). Alternatively, each fault segment could be analysed in a

separate model and resulting deformation be combined, providing a linear rheology is used. The model can, however, account for varying slip and rake along the single fault plane by specifying individual values at each node pair. At present the fault is not permitted to open, and whilst this is a realistic scenario for a fault, it would have negligible impact on the far-field postseismic deformation.

We have demonstrated the validity of this model for far-field coseismic and postseismic deformations. It is an improvement on existing models as it includes global spherical geometry, self-gravitation, and can be adapted to include 3D Earth structure. It will prove particularly useful for investigating earthquakes in regions that may have large lateral variations in Earth properties where a 1D Earth model cannot reproduce geodetic observations. Furthermore, the capability of *Abaqus* to model surface loading such as a changing ice load has already been established (e.g. van der Wal et al., 2010; Wu, 2004), and could

easily be incorporated into this model.


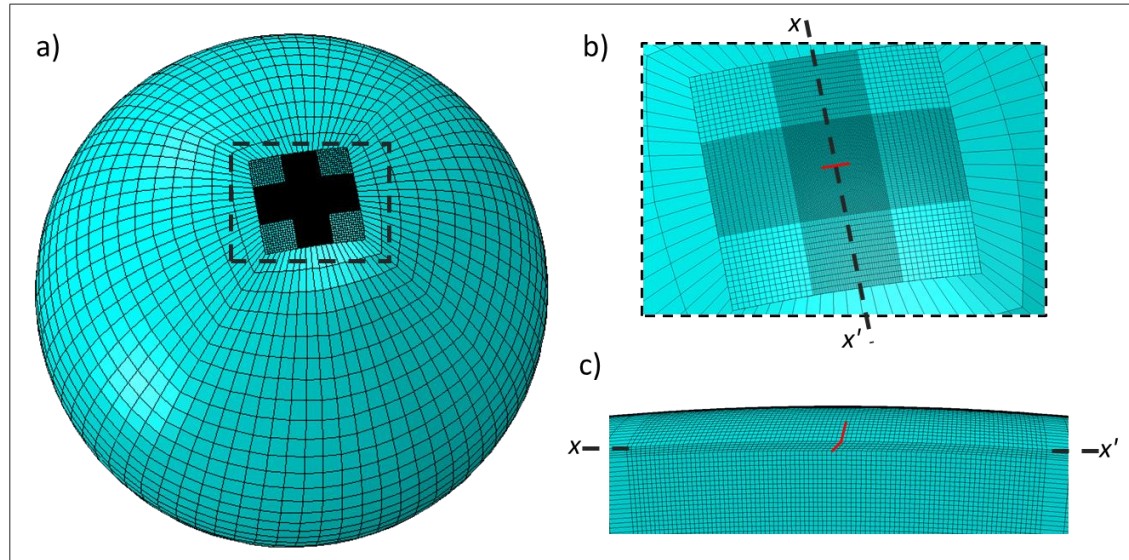

**Figure 1:** *Abaqus* mesh; a) full global mesh with high-resolution fault region surrounded by lower resolution elements; b) close up plan view of the fault region with fault marked in red and cross-section x-x' shown in c); c) cross-section x-x' (as marked in b)) through the fault region for 45° dipping fault with fault marked in red.

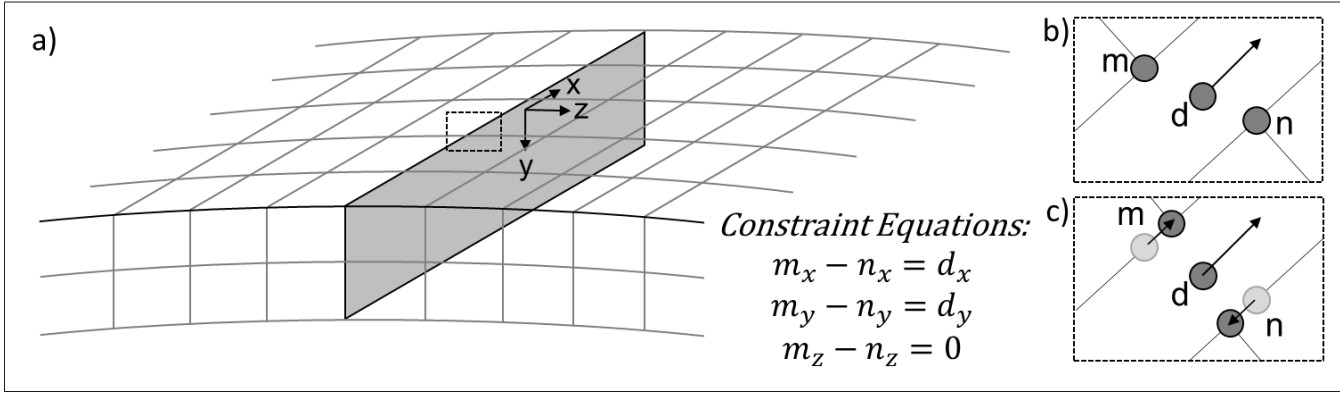

**Figure 2:** a) Diagram of the fault plane (shaded grey) in the *Abaqus* mesh with co-located node pairs and local coordinate system aligned with the fault plane; b) close up of a node pair (m, n) and dummy node (d) with displacement boundary condition applied in the x direction of the local coordinate system (for illustrative purposes only, the fault is not allowed to open in the z direction); c) constraint equations applied to the node pair results in displacement of nodes m and n in the x direction.

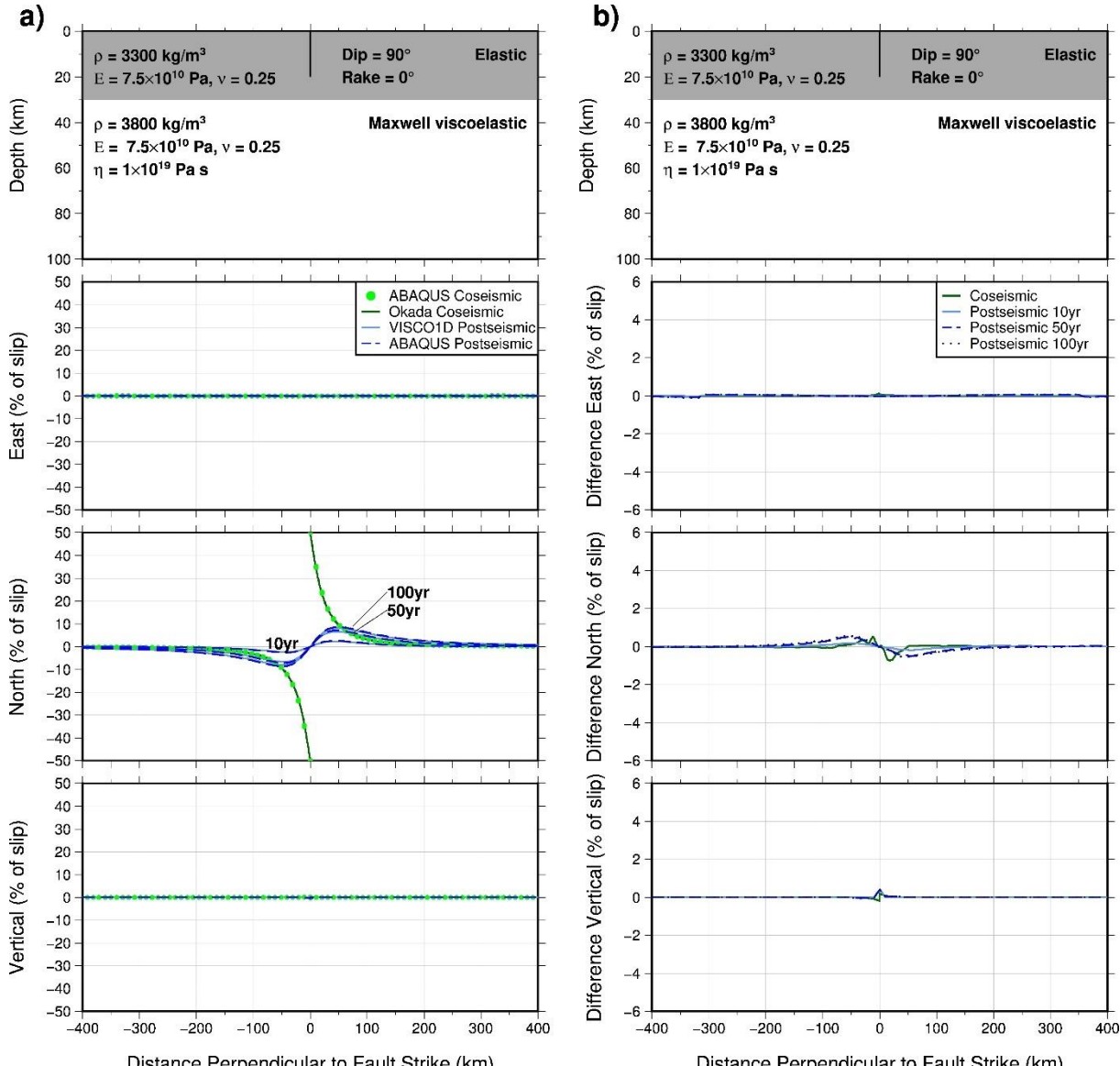

**Figure 3: Top panels of a) and b) show the fault dimensions and material properties of the upper 100 km of the model. Layers and material properties below 100 km depth are given in Table 2. a) Coseismic (green) and postseismic (blue) surface displacements in east, north and vertical directions in response to slip on a strike slip fault for a profile perpendicular to the fault strike. Results are calculated using the Okada analytical solution, *VISCO1D*, and *Abaqus* (see legend). Postseismic displacement is shown for times 10, 50, and 100 years after fault slip. Earth properties and fault dimension are given in top panel. Displacements are given as a percentage of the fault slip. b) Differences between the *Abaqus* coseismic displacement and the Okada analytical solution (green) and the *Abaqus* postseismic displacement and the *VISCO1D* displacement (blue). Differences are in terms of percentage of the fault slip.**

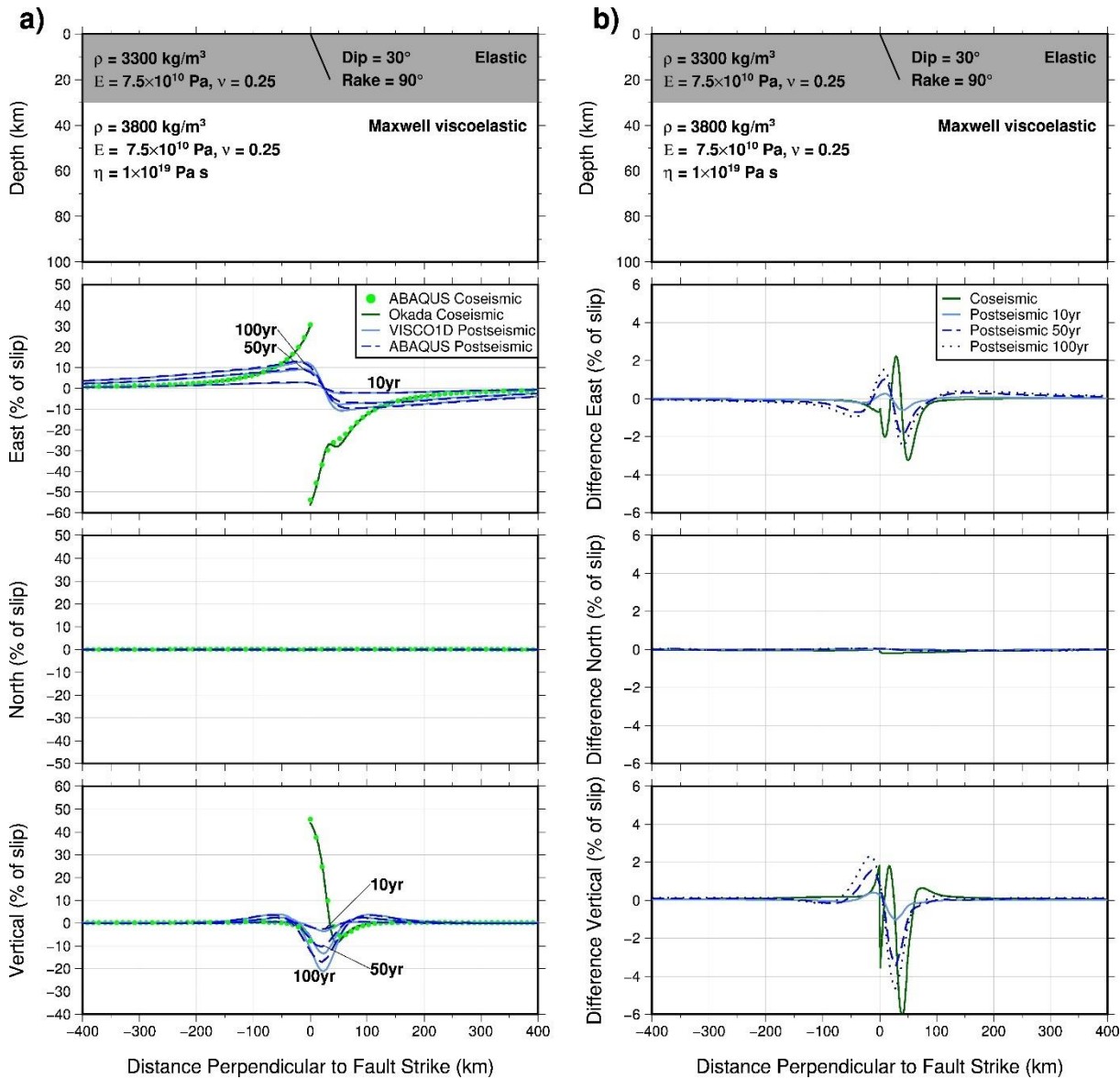


Figure 4: As for Figure 3 but for a 30° dipping reverse fault.


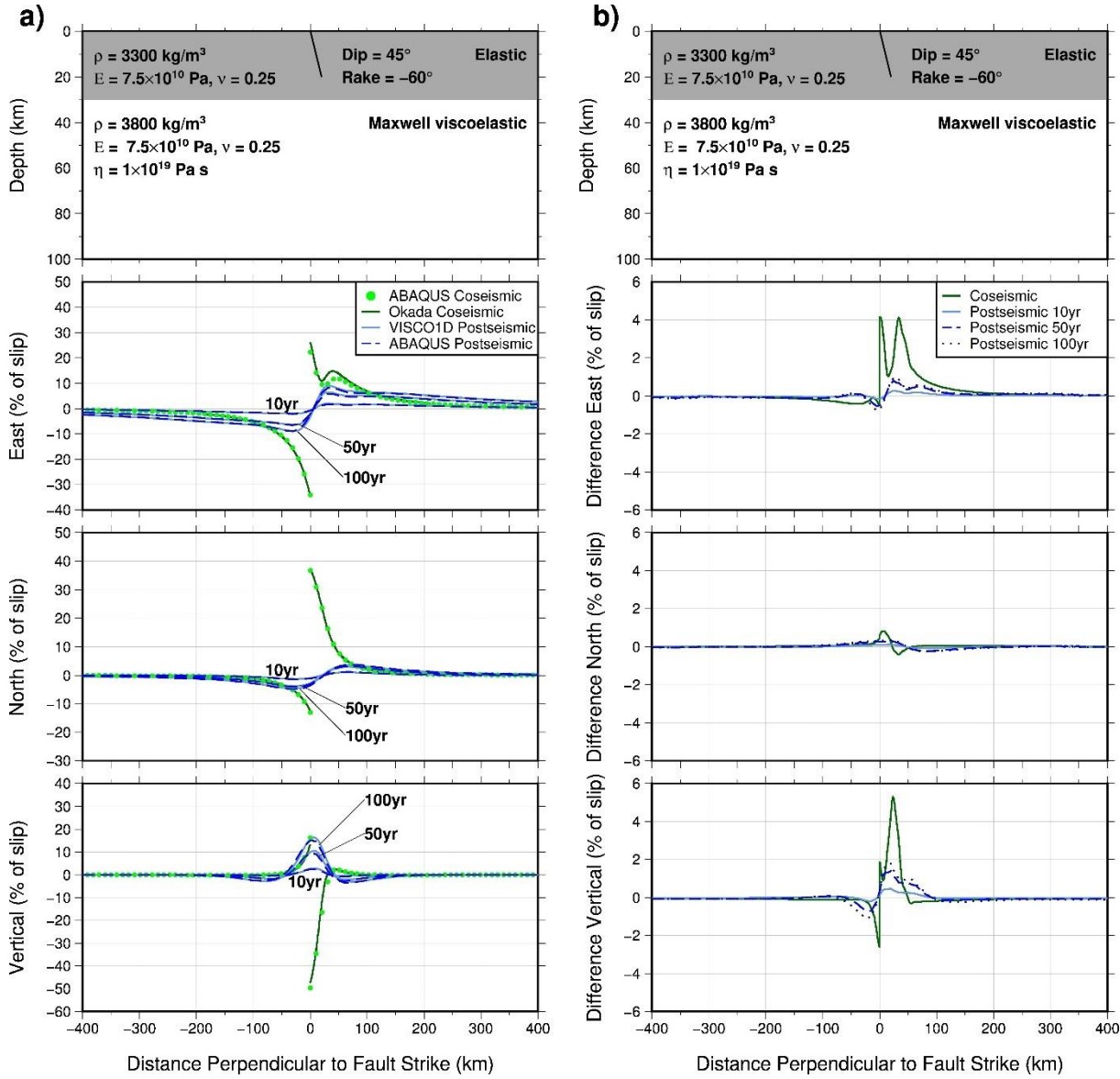

**Figure 5: As for Figure 3 but for a 45° dipping fault.**


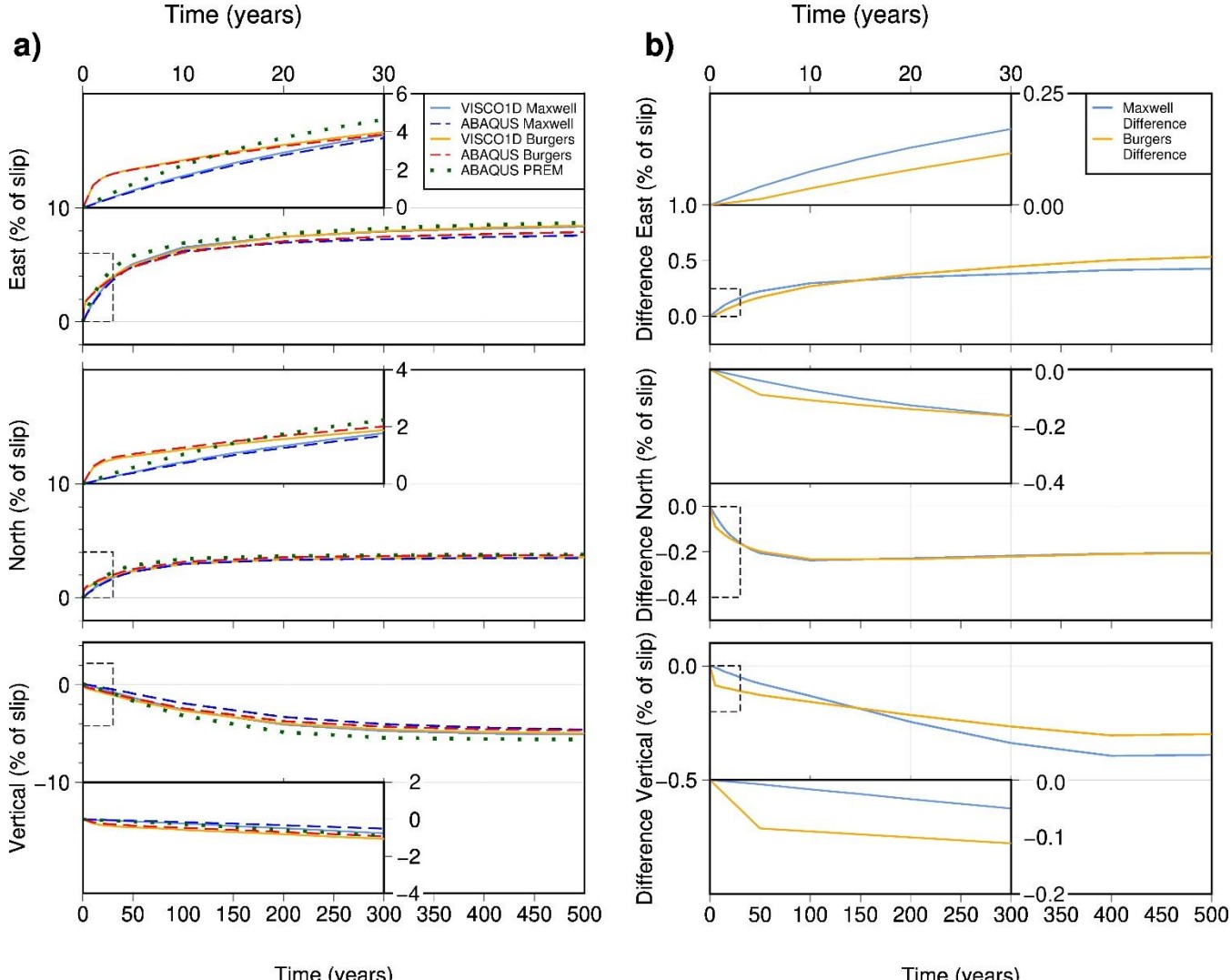

**Figure 6: a)** Postseismic surface displacement through time in the east, north and vertical directions at a location 100 km perpendicular to the fault in Figure 5. Predictions using two rheologies are shown: Maxwell rheology using *VISCO1D* and *Abaqus* ; and Burgers rheology using *VISCO1D* and *Abaqus* (see legend). Predictions from *Abaqus* using the PREM elastic properties are also shown (green dots). The main plot for each displacement direction shows 500 years of displacement with the first 30 years shown in detail in the insets. Displacements are given as a percentage of the fault slip. **b)** Difference between the *Abaqus* postseismic displacement and the *VISCO1D* displacement for Maxwell rheology (light blue line) and Burgers rheology (orange line). Positive indicates *Abaqus* has less displacement than *VISCO1D* and negative indicates *Abaqus* has more displacement than *VISCO1D*. Differences are in terms of percentage of the fault slip. Note different scales on the y-axis.

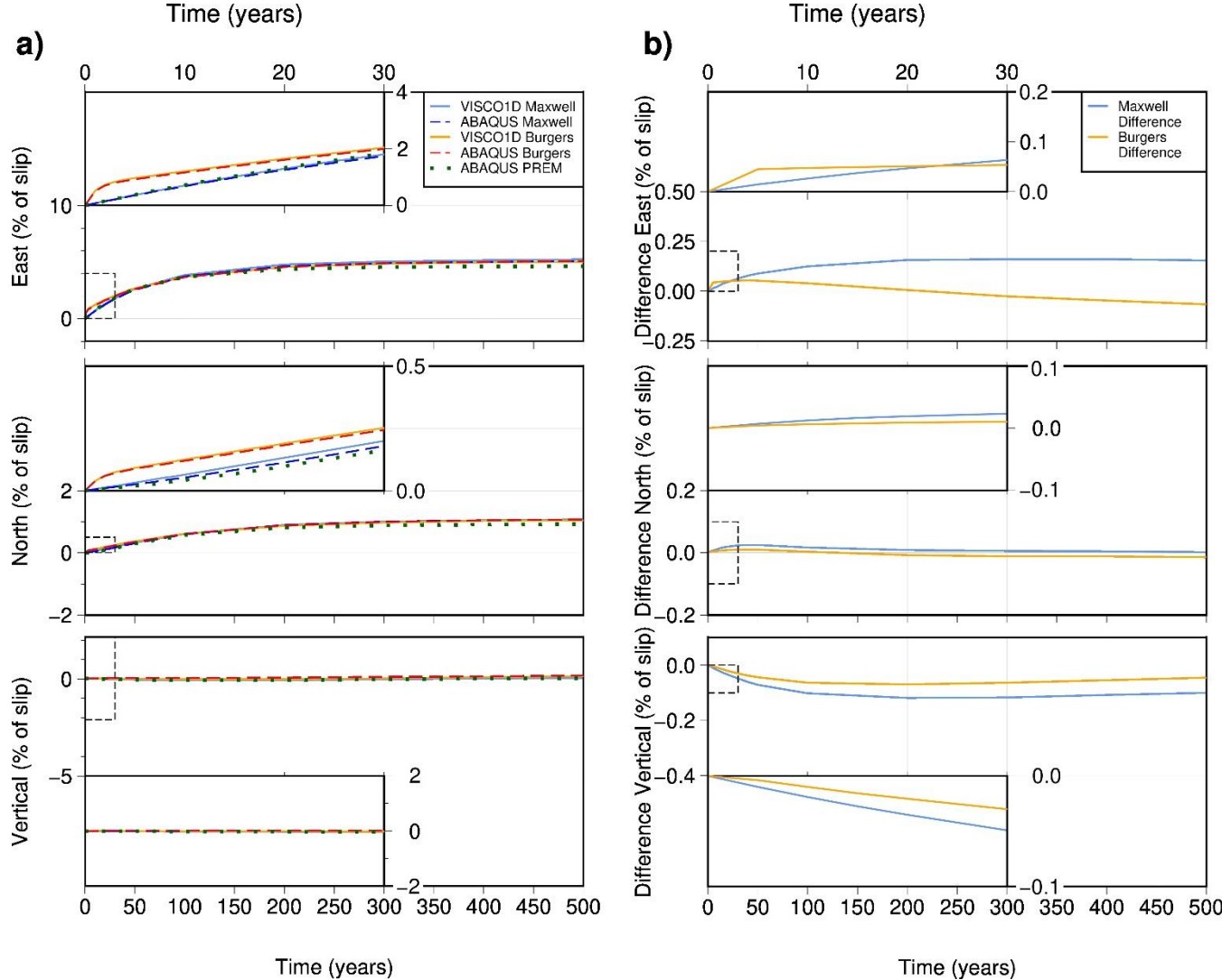

**Figure 7: As for Figure 6 but for a location 300 km perpendicular to the fault in Figure 5. Note different scales on the y-axis.**

**Table 1: Fault geometry used in the three benchmarking exercises. Note that test case 3 is run with 3 Earth models, see Table 2.**

| Test | Fault length (km) | Fault width (km) | Fault vertical depth (km) | Dip (°) | Strike (°) | Rake (°) | Slip (m) |
|---|---|---|---|---|---|---|---|
| 1. Strike-slip | 200 | 20 | 20 | 90 | 0 | 0 | 5 |
| 2. Reverse fault | 200 | 40 | 20 | 30 | 0 | 90 | 5 |
| 3. Normal oblique | 200 | 28.28 | 20 | 45 | 0 | -60 | 5 |

**Table 2: Earth model used in the *Abaqus* model for the benchmarking exercises with a Maxwell rheology and a Burgers rheology (transient viscosity given in brackets). The Maxwell viscosity profile is also used in conjunction with the PREM elastic structure for test case 3.**

| Layer Description | Depth of top of layer (km) | Layer thickness (km) | Density, $\rho$ (kg/m³) | Young's Modulus (Pa) | Poisson's Ratio, $v$ | Viscosity, $\eta$ (Pa s) | Vertical Element Resolution (km) |
|---|---|---|---|---|---|---|---|
| Elastic Lithosphere | 0 | 30 | 3300 | 7.50E+10 | 0.25 | Purely Elastic | 5 |
| Lower lithosphere/ Upper Mantle | 30 | 640 | 3800 | 7.50E+10 | 0.25 | 1.00E+19 (transient viscosity 1.00E+18) | 10 (from 30 to 400km depth) 50 (from 400 to 670km depth) |
| Lower Mantle | 670 | 2221 | 5000 | 1.10E+11 | 0.25 | 1.00E+21 | 500 |

## Appendix A: Mesh Resolution Test

In general, the higher the mesh resolution the more accurate the predictions of coseismic and postseismic displacement. However, increases in the mesh resolution in the near field of the fault quickly result in a prohibitively large number of elements in the model, and it becomes computationally too expensive to run. To verify that our choice of mesh resolution is sufficient, we set up the 30° reverse fault test with a coarser and a finer mesh resolution and compared the model-predicted displacement in the near- and far-field. Since the aim of our model is to calculate far-field postseismic deformation, the convergence of displacement at distances of 300 km from the fault is what determines the choice of mesh resolution, although results are given for the near-field for interest.

Table A1 shows the resolution of each mesh along with computation time for one iteration of the model. Note, this computation time is for parallel processing on 8 cores on a standard Linux workstation. Due to the way *Abaqus* licensing works, individual

set ups may be faster than the time quoted in Table A1 if users have access to more cores and licences. We term the mesh used
in the main body of the paper the "reference mesh".

**Table A1: Details of the meshes used in the sensitivity test.**

|  | Coarse Mesh | Reference Mesh | Fine Mesh |
|---|---|---|---|
| Lateral resolution of elements on the fault plane (km) | 20 | 10 | 5 |
| Depth resolution of elements on the fault plane (km) | 10 | 5 | 5 |
| Number of elements on the fault plane | 20 | 80 | 160 |
| Number of elements in the model | 161,366 | 417,309 | 1,250,590 |
| Computation time for one iteration | 45 minutes | 8 hours | 84 hours |

We evaluate overall displacement results in terms of differences in percentage of fault slip to be consistent with Figs. 3-7. The
key results are summarised in Table A2. Comparing results for the reference mesh with the coarse mesh shows that near-field
coseismic differences peak at 4.5% of the fault slip and the maximum difference in the postseismic deformation after 100 years
is 3.1%. When comparing the reference mesh with the fine mesh, this reduces to 1.5% for coseismic differences and 1.2% for
postseismic differences.

In the far-field however, differences between the coarse mesh and reference mesh are small, peaking at 0.2% of the fault slip
after 100 years of postseismic deformation. Far-field differences between the reference mesh and the fine mesh are negligible.
Since the focus of our model is the far-field, these results might suggest that any of the mesh resolutions tested would be
sufficient. However, in terms of absolute displacement we see an improvement from differences of 5-9 mm between the coarse
mesh and the reference mesh, to 0.1-1.6 mm for differences between the reference and fine meshes. This demonstrates that
results have converged at the resolution of our chosen mesh to be less than the magnitude of uncertainty associated with GPS-
observed postseismic deformation (e.g., Freed et al., 2006; Wang and Fialko, 2018). We therefore conclude that the chosen
mesh resolution is more than fit for purpose, with only small improvements to near-field displacement to be gained by using a
finer mesh.


**Table A2: Differences in displacement (expressed as percentage of fault slip) between the different meshes in Table A1. In brackets is the direction of displacement where the maximum difference is observed.**

| | | Reference Mesh minus Coarse Mesh | Fine Mesh minus Reference Mesh |
|---|---|---|---|
| Near-field (at the fault location) | Coseismic | 4.5% (vertical) | 1.5% (east) |
| | Postseismic after 100 years | 3.1% (east) | 1.2% (east) |
| Far-field (300 km from the fault) | Coseismic | 0.1% (east) | 0% (east) |
| | Postseismic after 100 years | 0.2% (east) | 0.03% (east) |

## Appendix B

This section shows surface displacement for the *Abaqus* model-predicted coseismic and postseismic displacement for each of the three benchmarking tests with Maxwell rheology, and for test case 3 using the elastic properties from PREM.


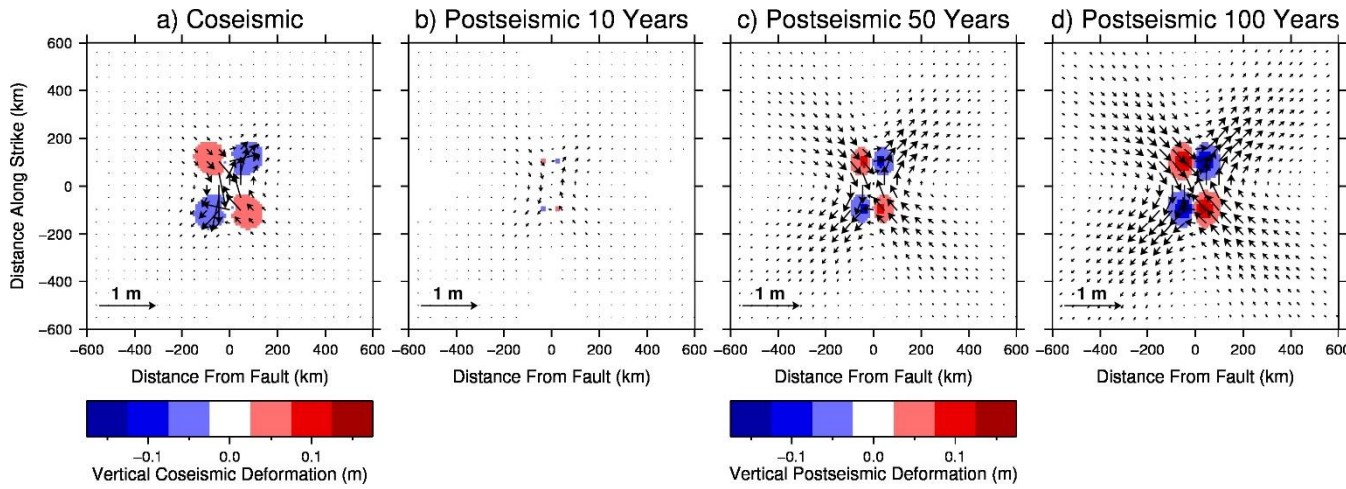

**Figure B1: Model-predicted surface displacement in response to slip on a strike slip fault. Background colours show vertical displacement and arrows show horizontal displacement for: a) Coseismic displacement; b), c) and d) postseismic deformation at 10, 50, and 100 years after fault slip respectively.**


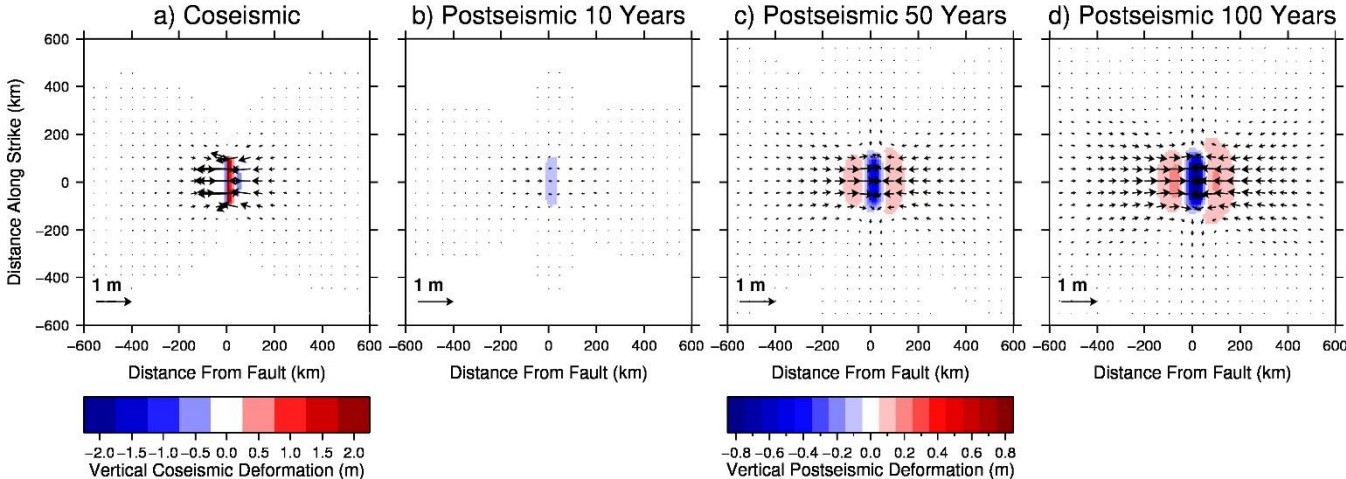

**Figure B2:** As for Figure B1 but for a 30° dipping reverse fault. Note the different colour scales for coseismic and postseismic displacement.

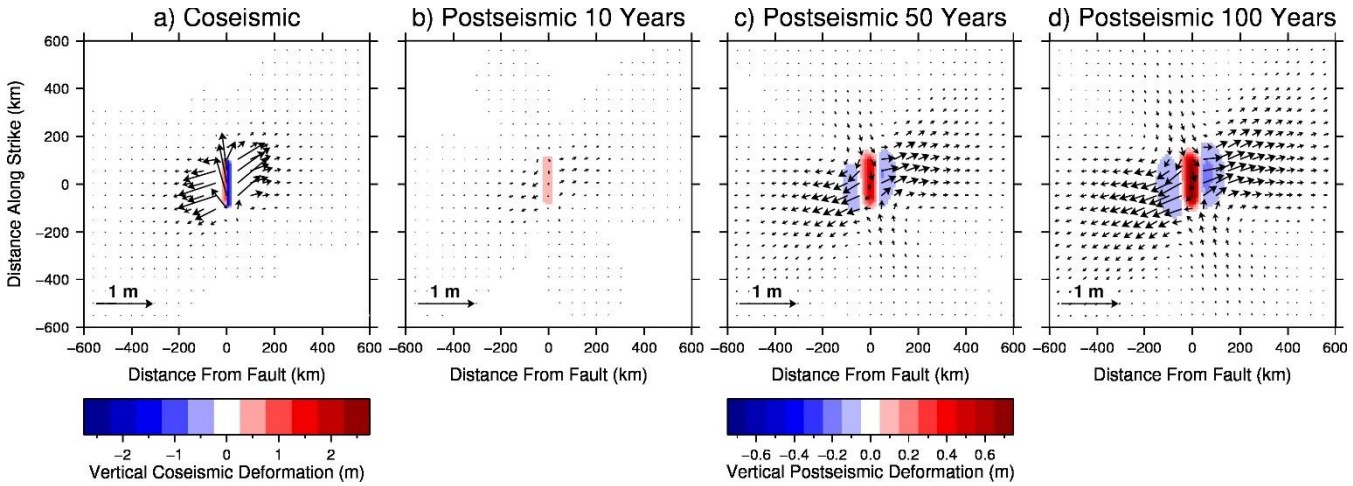


**Figure B3:** As for Figure B1 but for a 45° dipping fault. Note the different colour scales for coseismic and postseismic displacement.

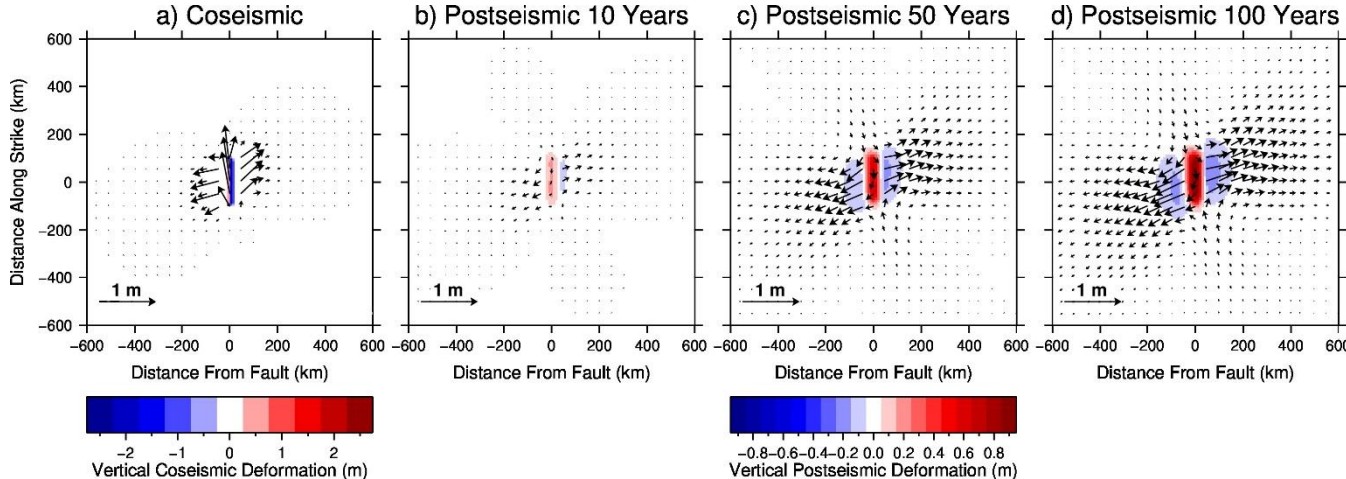

**Figure B4: As for Figure B3, a 45° dipping fault, but using the PREM Earth model. Note the different colour scales for coseismic and postseismic displacement.**


*Code Availability. Abaqus* is a commercial software and can be purchased from the developer (https://www.3ds.com/products-services/simulia/products/abaqus/). We have used version 2018 in this study, but the methods and functionality used are available in older versions of the software as well. *Abaqus* input files for all five models presented in this paper are available

at    https://github.com/ganield/ABAQUS_Postseismic_model/releases/tag/v2.0,    and    archived    on    Zenodo: https://doi.org/10.5281/zenodo.5897863 (Nield, 2022). Instructions on how to run the files are also available at this link. *VISCO1D* version 3 is available for download at: https://www.usgs.gov/software/visco1d. The *VISCO1D* input files and Earth model files used in the experiments presented in this paper are included in the github repository, along with some instructions on how to run the code.


*Author Contributions.* GN developed the model. MK conceived the study and consulted in the model implementation. RS contributed meshing code for *Abaqus* and consulted in the model development. BB contributed the self-gravitational part of the code. All authors contributed to the manuscript.

*Competing Interests.* None.

*Acknowledgements.* GN is supported by ARC Discovery Project DP170100224. We are grateful to Fred Pollitz for making *VISCO1D* freely available and for his assistance with using the software. We also thank Tim Masterlark for his advice on implementing fault slip in *Abaqus* and Wouter van der Wal for guidance on constructing *Abaqus* models. The handling editor

Lutz Gross and two anonymous reviewers are gratefully acknowledged for their constructive comments.

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
