# Peer review of "A global, spherical, finite-element model for postseismic deformation using *Abaqus"

_Geoscientific Model Development, 2020_

## Referee Comment (RC1) · Anonymous Referee #1 · 18 Aug 2020

The authors used ABAQUS —a commercial finite-element software package— to simulate postseismic deformation on the self-gravitating earth model. They have benchmarked their results for both coseismic and postseismic deformations with semianalytical solutions.

The article's subject matter is interesting and relevant to the journal of Geoscientific Model Development (GMD). The results for the given examples look excellent. I have the impression that the overall content of the article could be improved. I have a few concerns.

- The authors used the finite-element method, which is also clearly implied from the

title. However, I do not see any related finite element formulations. I expect at least the strong and weak forms of the governing equations with necessary boundary conditions so that the work is entirely reproducible. For example, implementing full gravity and solid-fluid coupling is known to be challenging for global problems. I am curious about how those aspects are implemented. In my view, a proper section for appropriate formulations would make this article complete.

- The most basic and widely used Earth model is the Preliminary reference Earth model (PREM, Dziewonski, A. M. & Anderson, D. L., 1981). I wonder why authors chose to use a simple three-layered model instead of the PREM. Furthermore, they mention in the abstract "the model can be easily adapted to include different rheological models and lateral variations". In this context, at least one example with the lateral variation of viscosity (e.g., Latychev et al., 2005) would be interesting.

- Although not explained in the article, it seems that the mesh contains the nonconforming elements when transitioning from course to fine elements as shown in Figure 1. But then in Section "2.1 Model Geometry and Mesh" the authors mention, "The element type used is an 8-node linear brick element"! How is it possible to use an 8-node brick element for nonconforming elements? Do you use a discontinuous Galerkin method? Please clarify.

- Authors have frequently used the term "flat earth." I think "homogeneous halfspace" or "layered halfspace" is probably a more appropriate term.

Given the above comments, I would recommend this article for a moderate to major revision.

Minor comments follow. In the comments below, P refers to the page number, and L refers to the line number.

P4L29: "..on the likely Earth structure..." What do you mean by "likely Earth structure"?

P2L34: "flat-Earth." "Homogeneous halfspace" or "layered halfspace"?

[Figure]

P3L92: "a fault plane within the mesh." Given that you use the brick elements, accommodating the realistic and complex faults may be very difficult with this approach. Alternatively, one can use the so-called moment-density tensor approach.

P3L96: "...using surface-to-surface tie constraints..." Please write appropriate equations for these constraints.

P4 Section 2.3: These boundary conditions are best to be represented by appropriate equations!

P6L165: "...as the fault is not allowed to open." Realistic faults may have some opening as well. How do you accommodate that kind of scenario?

P6L181: "500 km." Given that the total depth of the model is 670 km, how does this large element behave?

P7L185: "... simple Earth structure". Why not use a more common Earth model PREM?

P7 Section 4.2 Coseismic Results Can you show the snapshots of the surface displacement?

P7 Section 4.3 Postseismic Results Can you show the snapshots of the surface displacement at selected time steps?

P7L214: "...less coseimic dispalcemetn from the ABAQUS..." What is the reason for less coseismic displacement for ABAQUS?

P9L251: "...an approximation of all the fault planes into a single geometry would be required.." I don't think this is a reliable way. A better alternative is to use the moment-density tensor approach.

Figure 1: This figure may be sharper and better in black and white.

Figures 3-5: Showing the depth only to 100 km is confusing. Either show the full depth

or explain it in the captions.

Figures 3-7: Figures look low in quality. It may be better to save those figures in vector graphics, if possible.

Finally, the following references are worth citing. References: Al-Attar, D. & Tromp, J., 2014: Sensitivity kernels for viscoelastic loading based on adjoint methods Geophysical Journal International, Oxford University Press, 196, 34-77

Dziewonski, A. M. & Anderson, D. L., 1981: Preliminary reference Earth model Physics of the Earth and Planetary Interiors, 25, 297-356

Latychev, K.; Mitrovica, J. X.; Tromp, J.; Tamisiea, M. E.; Komatitsch, D. & Christara, C. C., 2005: Glacial isostatic adjustment on 3-D Earth models: a finite-volume formulation Geophysical Journal International, Blackwell Science Ltd, 161, 421-444

Zhong, S.; Paulson, A. & Wahr, J., 2003: Three-dimensional finite-element modelling of Earth's viscoelastic deformation: effects of lateral variations in lithospheric thickness Geophysical Journal International, 155, 679-695

Best regards,

---

## Referee Comment (RC2) · Anonymous Referee #2 · 22 Sep 2020

Review of "A global, spherical, finite-element model for postseismic deformation using ABAQUS" by Nield and co-authors.

The manuscript represents an implementation of postseismic viscoelastic relaxation problems in a widely used finite-element commercial package. The study addresses common problems associated with meshing the domain, which is difficult around faults, and benchmarks the results against semi-analytic solutions attained with another widely used, but open-source, package. The study is accompanied with supplementary material that allows the community to reproduce and expand on these results quickly.

[Figure]

The study makes a number of simplifying assumptions about the rheology of the Earth that permits direct comparison with the semi-analytic code visco1d. However, once the code is benchmarked, these assumptions should be relaxed and more realistic constitutive laws that include a power-law stress/strain-rate relationship at steady state and a similar power-law constitutive behavior for transient creep - all compatible with laboratory observation of olivine creep - should be implemented and described. More realistic distributions of physical properties associated with thermal activation of viscoelastic flow in a realistic thermal field should follow.

A remaining issue is the meshing around more complex fault assembly that include multiple surfaces is not included in the model. As many earthquakes are now imaged to such a level of accuracy that these details are often well constrained, including complex fault geometry would be a relevant addition.

Finally, the iterative procedure to include self-gravity should be replaced by directly solving the appropriate equations based on advection of pre-stress.

I follow with a few detailed remarks.

55: An example of finite-element modeling of post-seismic relaxation with a spherical geometry is

Agata, R., Barbot, S.D., Fujita, K., Hyodo, M., Iinuma, T., Nakata, R., Ichimura, T. and Hori, T., 2019. Rapid mantle flow with power-law creep explains deformation after the 2011 Tohoku mega-quake. Âă Nature communications, Âă 10(1), pp.1-11.

105: Since it seems so easy to add more realistic rheology with the method, it should actually be done in this study. More realistic rheology involves a power-law stress/strain-rate relationship, see

Hirth, G. and Kohlstedt, D.L., 2003. Rheology of the Upper Mantle and the Mantle Wedge: A View from the Experimentalists: Inside the Subduction Factory, v. 138.

Karato, S.I. and Wu, P., 1993. Rheology of the upper mantle: A synthesis. Science,

260(5109), pp.771-778.

Recent development include the inclusion of transient creep compatible with nonlinear steady-state creep:

Masuti, S., Barbot, S.D., Karato, S.I., Feng, L. and Banerjee, P., 2016. Upper-mantle water stratification inferred from observations of the 2012 Indian Ocean earthquake. Nature, 538(7625), pp.373-377.

Inclusion of realistic rheology seems more important and relevant than including self-gravitation.

130: It is unfortunate that Abaqus cannot simply solve the appropriate governing equations for self-gravitation and that these iterations are necessary. How can that be improved? Is there a way to solve a user-defined set of equations? Are the governing equations with self-gravitation not readily included in Abaqus? How is advection of pre-stress included?

180: The horizontal and vertical resolutions of the mesh seem inadequate to resolve the near field. The fault is 200x20 km and the mesh size around it is 10x5 km, representing just 20x4 mesh elements along the fault. It is actually surprising that the numerical result match the analytic solution so well with such a coarse mesh. This is perhaps an area of improvement.

215: We need to see a convergence test in terms of mesh resolution for these cases. This may not necessitate more figures, but this needs to be discussed. I suspect that the resolution of the mesh in the near field can be improved, with valuable gains on the misfit. I suspect that the discrepancies that accumulate at long period during postseismic relaxation may be reduced with a more appropriate mesh in the near field.

250: If linear rheology models are assumed, several simulations can be run with separate parts of a complex fault geometry model - each fault at a time - and the results subsequently combined.

---

## Author Comment (AC2) · 28 Sep 2021

We thank the reviewer for these suggestions and have responded to the detailed comments in line below.

-Review of "A global, spherical, finite-element model for postseismic deformation using ABAQUS" by Nield and co-authors.

-The manuscript represents an implementation of postseismic viscoelastic relaxation problems in a widely used finite-element commercial package. The study addresses common problems associated with meshing the domain, which is difficult around

faults, and benchmarks the results against semi-analytic solutions attained with an-other widely used, but open-source, package. The study is accompanied with supple-mentary material that allows the community to reproduce and expand on these results quickly.

-The study makes a number of simplifying assumptions about the rheology of the Earth that permits direct comparison with the semi-analytic code visco1d. However, once the code is benchmarked, these assumptions should be relaxed and more realistic consti-tutive laws that include a power-law stress/strain-rate relationship at steady state and a similar power-law constitutive behavior for transient creep - all compatible with lab-oratory observation of olivine creep - should be implemented and described. More realistic distributions of physical properties associated with thermal activation of vis-coelastic flow in a realistic thermal field should follow.

Power-law and transient power-law rheology have not been included in this study as the primary aim is to benchmark coseismic and postseismic displacement results against those produced by existing models with linear rheology. The implementation in ABAQUS of the rheologies mentioned by the reviewer is straight forward and has been done in other studies as mentioned on line 108. We will add more detail and references such as those suggested by the reviewer to section 2.2 to expand on this. However, using our model with more complex rheology will be the subject of future work and we feel that this is outside of the scope of our benchmarking study.

-A remaining issue is the meshing around more complex fault assembly that include multiple surfaces is not included in the model. As many earthquakes are now imaged to such a level of accuracy that these details are often well constrained, including complex fault geometry would be a relevant addition.

We agree this remains a limitation of the model due to the difficulties in constructing a mesh around a complex fault structure with brick elements. However, we are focusing on the far-field postseismic displacement which is less sensitive to simplifications made

to the fault geometry than near-field displacement (Khazaradze et al., 2002, Tregoning et al., 2013, Zhou et al. 2012). Representing complex fault geometry with a single plane geometry can provide a useful way of modelling far-field postseismic deformation. This method has been used by Takeuchi and Fialko (2013), and we will include a reference to this study on line 251. When applied to case studies, fault and slip properties in the model can be adjusted so that model output matches observations of coseismic displacements which provides further confidence in modelled far-field deformations (e.g. Sun et al., 2018).

-Finally, the iterative procedure to include self-gravity should be replaced by directly solving the appropriate equations based on advection of pre-stress.

This approach is not possible for a spherical model in ABAQUS, please also see more detailed response to comment below.

-I follow with a few detailed remarks.

-55: An example of finite-element modeling of post-seismic relaxation with a spherical geometry is Agata, R., Barbot, S.D., Fujita, K., Hyodo, M., Iinuma, T., Nakata, R., Ichimura, T. and Hori, T., 2019. Rapid mantle flow with power-law creep explains deformation after the 2011 Tohoku mega-quake.ÂaNature communications, Â ËŸ a10(1), pp.1-11. ËŸ

This additional reference will be included.

-105: Since it seems so easy to add more realistic rheology with the method, it should actually be done in this study. More realistic rheology involves a power-law stress/strain-rate relationship, see

Hirth, G. and Kohlstedt, D.L., 2003. Rheology of the Upper Mantle and the Mantle Wedge: A View from the Experimentalists: Inside the Subduction Factory, v. 138. Karato, S.I. and Wu, P., 1993. Rheology of the upper mantle: A synthesis. Science, C2 GMDD Interactive comment Printer-friendly version Discussion paper 260(5109),

pp.771-778. Recent development include the inclusion of transient creep compatible with nonlinear steady-state creep: Masuti, S., Barbot, S.D., Karato, S.I., Feng, L. and Banerjee, P., 2016. Upper-mantle water stratification inferred from observations of the 2012 Indian Ocean earthquake. Nature, 538(7625), pp.373-377.

-Inclusion of realistic rheology seems more important and relevant than including self-gravitation.

We feel this is outside the scope of the benchmarking study, please also refer to our earlier comment.

-130: It is unfortunate that Abaqus cannot simply solve the appropriate governing equations for self-gravitation and that these iterations are necessary. How can that be improved? Is there a way to solve a user-defined set of equations? Are the governing equations with self-gravitation not readily included in Abaqus? How is advection of pre-stress included?

The governing equations solved in ABAQUS cannot be changed. A gravity load can be included directly within ABAQUS as a uniform acceleration in one fixed direction, therefore it is not easily applied to a spherical model. We choose instead to use the iterative approach as this has been shown by others (Wu, 2004) to correctly represent self-gravitation for a spherical viscoelastic Earth.

Advection of pre-stress is included via the elastic foundations described in section 2.3. We will add additional text to this section to clarify.

-180: The horizontal and vertical resolutions of the mesh seem inadequate to resolve the near field. The fault is 200x20 km and the mesh size around it is 10x5 km, representing just 20x4 mesh elements along the fault. It is actually surprising that the numerical result match the analytic solution so well with such a coarse mesh. This is perhaps an area of improvement.

We are focusing on the far-field postseismic deformation within a global setting, which

requires a mesh with a very large number of elements. It is therefore computationally expensive to have a very high-resolution mesh in the near-field and we are forced to trade off the mesh resolution (and hence more accurate near-field results) against computation time. To provide further justification for our choice of mesh resolution we will perform extra sensitivity tests for one of the fault cases and discuss the results in the text.

-215: We need to see a convergence test in terms of mesh resolution for these cases. This may not necessitate more figures, but this needs to be discussed. I suspect that the resolution of the mesh in the near field can be improved, with valuable gains on the misfit. I suspect that the discrepancies that accumulate at long period during postseismic relaxation may be reduced with a more appropriate mesh in the near field.

As per our response to the previous comment, we will perform extra sensitivity tests for mesh resolution and discuss the results in the text. We will pay particular attention to improvements in misfit and the trade of in computation time.

-250: If linear rheology models are assumed, several simulations can be run with separate parts of a complex fault geometry model - each fault at a time - and the results subsequently combined.

This is an approach that could work for modelling complex faults with linear rheology, we will amend the text to include this point.

————————————- Additional References:

Khazaradze, G., Wang, K., Klotz, J., Hu, Y., and He, J., Prolonged post-seismic deformation of the 1960 great Chile earthquake and implications for mantle rheology, Geophys. Res. Lett., 29( 22), 2050, doi:10.1029/2002GL015986, 2002.

Sun, T., Wang, K. & He, J., 2018. Crustal Deformation Following Great Subduction Earthquakes Controlled by Earthquake Size and Mantle Rheology, Journal of Geophysical Research: Solid Earth, 123, 5323-5345.

Tregoning, P., Burgette, R., McClusky, S. C., Lejeune, S., Watson, C. S., and McQueen, H. (2013), A decade of horizontal deformation from great earthquakes, J. Geophys. Res. Solid Earth, 118, 2371– 2381, doi:10.1002/jgrb.50154.

Zhou, X., Sun, W., Zhao, B., Fu, G., Dong, J., and Nie, Z. (2012), Geodetic observations detecting coseismic displacements and gravity changes caused by the Mw = 9.0 Tohoku-Oki earthquake, J. Geophys. Res., 117, B05408, doi:10.1029/2011JB008849.

---

## Author Response (AR1)

**Final Response**

Dear Editor,

Firstly, we wish to thank you and the reviewers for your continued patience throughout this review process and for agreeing to extend the deadline for such a long period of time. This request was due to myself taking an extended and unexpectedly early maternity leave and as a co-author team we decided it would be better to wait until my return to work to respond to the reviews.

We are grateful to both reviewers for their constructive comments on our manuscript. Please find in this final response a point-by-point reply to all reviewer comments (red text), including the changes we have made to the manuscript (marked in italics). The revised manuscript is attached below the response to the reviewers.

Kind Regards

Grace Nield

**Response to Reviewer 1**

The authors used ABAQUS —a commercial finite-element software package— to simulate postseismic deformation on the self-gravitating earth model. They have benchmarked their results for both coseismic and postseismic deformations with semianalytical solutions.

The article's subject matter is interesting and relevant to the journal of Geoscientific Model Development (GMD). The results for the given examples look excellent. I have the impression that the overall content of the article could be improved. I have a few concerns.

We thank the reviewer for these helpful comments and have responded in line below.

- The authors used the finite-element method, which is also clearly implied from the title. However, I do not see any related finite element formulations. I expect at least the strong and weak forms of the governing equations with necessary boundary conditions so that the work is entirely reproducible. For example, implementing full gravity and solid-fluid coupling is known to be challenging for global problems. I am curious about how those aspects are implemented. In my view, a proper section for appropriate formulations would make this article complete.

We have not included the governing equations for the finite element formulation as this part of the study is not new work. The model is based on the finite element formulation of Wu (2004) and equations therein, as referenced on line 84. Instead, we focus the paper on describing the new aspects of this model – incorporating a fault plane and prescribing slip. To make the work more reproducible we have added further references to Wu (2004) through Section 2.3 as follows:

*We follow the approach described in Section 4.1 of Wu (2004) and apply elastic foundations (ABAQUS keyword \*Foundation) to each layer boundary with a material density contrast occurring across it (including the surface and core-mantle-boundary). This means that advection of pre-stress is included and takes care of the restoring forces of buoyancy neglected in a conventional finite-element model (Wu, 2004, equation 3). The elastic foundations have a stiffness equal to the difference in density multiplied by gravitational acceleration (see Wu (2004) equations 12a,b,c).*

**- The most basic and widely used Earth model is the Preliminary reference Earth model (PREM, Dziewonski, A. M. & Anderson, D. L., 1981). I wonder why authors chose to use a simple three-layered model instead of the PREM. Furthermore, they mention in the abstract "the model can be easily adapted to include different rheological models and lateral variations". In this context, at least one example with the lateral variation of viscosity (e.g., Latychev et al., 2005) would be interesting.**

We chose the same simple three-layer structure for this benchmarking exercise as that used by Pollitz (1997) (line 186). Since we were undertaking some benchmarking tests with the same fault geometry as those in Pollitz (1997) it was essential to keep the Earth model consistent with that study. Using the Preliminary Reference Earth Model (PREM) instead would not add anything to the results in our opinion, and only complicate reproducibility of the results for others.

We have added a sentence to new line 210 to explain:

*We use a simple Earth structure rather than the Preliminary Reference Earth Structure (PREM, Dziewonski & Anderson, 1981) to ensure our results are consistent with Pollitz (1997).*

Our study does not include lateral variations in viscosity as the primary aim was to benchmark the model against other existing models and there is no open-source spherical model that we are aware of with which to benchmark these results. Using the model for case studies that require lateral variations in viscosity will be the subject of future work.

**- Although not explained in the article, it seems that the mesh contains the nonconforming elements when transitioning from course to fine elements as shown in Figure 1. But then in Section "2.1 Model Geometry and Mesh" the authors mention, "The element type used is an 8-node linear brick element"! How is it possible to use an 8- node brick element for nonconforming elements? Do you use a discontinuous Galerkin method? Please clarify.**

We thank the reviewer for bringing to our attention this lack of detail. ABAQUS provides a useful way to join together meshes of different resolutions to aid mesh refinement problems. The two separate meshes are joined together by a "tie constraint" on a surface where they have non-conforming elements relative to each other. Using a tie constraint ensures there is no relative movement between the surfaces and that displacement and stress are continuous through the boundaries. Nodes on one mesh are tied to nodes on the other mesh. Tie coefficients are generated and used to interpolate quantities from nodes on one side of the mesh to nodes on the other side of the mesh. We have expanded the text in Section 2.1 to include further description of this method as below:

*The two parts are then tied together (ABAQUS keyword \*Tie) using surface-to-surface tie constraints. This means that although the two separate meshes have non-conforming elements relative to each other, the tie constraints ensure there is no relative movement between the surfaces and that displacement and stress are continuous through the boundaries. Tie coefficients are generated and used to interpolate quantities from nodes on one side of the mesh to nodes on the other side of the mesh.*

**- Authors have frequently used the term "flat earth." I think "homogeneous halfspace" or "layered halfspace" is probably a more appropriate term.**

We will change this term to "layered halfspace" but will also note on first use that it is also referred to as "flat-Earth" in other literature (e.g. Wu, 2004) so as not to cause confusion to the reader.

**Given the above comments, I would recommend this article for a moderate to major revision. Minor comments follow. In the comments below, P refers to the page number, and L refers to the line number.**

**P4L29: "..on the likely Earth structure..." What do you mean by "likely Earth structure"?**

We mean Earth structure inferred by the model. We will change this to "inferred Earth structure".

**P2L34: "flat-Earth." "Homogeneous halfspace" or "layered halfspace"?**

We will change this term to "layered halfspace".

**P3L92: "a fault plane within the mesh." Given that you use the brick elements, accommodating the realistic and complex faults may be very difficult with this approach. Alternatively, one can use the so-called moment-density tensor approach.**

We acknowledge that a limitation of the model is that it is currently restricted to a single fault plane, due to the difficulties in constructing a mesh around a complex fault structure, as discussed on line 246. The moment-density tensor approach suggested by the reviewer is an alternative method of representing a fault within a spectral-element mesh (e.g. Gharti et al. 2019) whereby the mesh geometry is not required to conform to the geometry of the fault plane. However, our current knowledge is that this method cannot be implemented in a finite-element mesh in ABAQUS due to the restrictions of defining loads and forces on elements or surfaces within the mesh. The focus of our study is far-field deformation which is not sensitive to the details of the fault plane and slip distribution (Khazaradze et al., 2002, Tregoning et al., 2013, Zhou et al. 2012), rather it is the overall moment magnitude that is important, which can be represented on a single plane. We have included additional text in the discussion as follows:

*In the case of a fault inversion that suggests multiple fault segments (e.g. Ye et al., 2014), an approximation of all the fault planes into a single geometry could still provide a realistic far-field estimate of postseismic deformation (e.g. Takeuchi and Fialko, 2013), particularly if the fault geometry and slip are adjusted so that model output matches observations of coseismic displacement (e.g. Sun et al., 2018). Far-field postseismic deformation is less sensitive to simplifications made to the fault geometry and slip*

*distribution than near-field deformation (Khazaradze et al., 2002, Tregoning et al., 2013, Zhou et al. 2012).*

**P3L96: "...using surface-to-surface tie constraints. . ." Please write appropriate equations for these constraints.**

The equations used by ABAQUS to define the tie constraints are integrated into the software so it would not be appropriate to reproduce them in this paper. The information we have included with regards to ABAQUS key words and input files is sufficient to allow other to use these methods.

**P4 Section 2.3: These boundary conditions are best to be represented by appropriate equations!**

We will include a more specific reference to the Wu (2004) study that this model is based on so that the reader can refer to the original source for the equations. Please also refer back to our response to the earlier comment.

**P6L165: "...as the fault is not allowed to open." Realistic faults may have some opening as well. How do you accommodate that kind of scenario?**

At present we do not accommodate opening faults although we appreciate this is a realistic fault scenario. We recognise that this is a limitation of our model and have included a further comment in the discussion section (new line 279) to acknowledge this.

> *At present the fault is not permitted to open, and whilst this is a realistic scenario for a fault, it would have negligible impact on the far-field postseismic deformation.*

**P6L181: "500 km." Given that the total depth of the model is 670 km, how does this large element behave?**

The total depth of the model is 2891 km – from the surface down to the lower mantle-core boundary. The elements of 500 km size are present only within the lower mantle layer and the global mesh that surrounds the region of refined mesh where the fault lies. We ensure that the majority of coseismic and postseismic displacement occurs in the inner part of the model where elements are much smaller, so that the large elements should be deforming by only a negligible amount.

**P7L185: "... simple Earth structure". Why not use a more common Earth model PREM?**

Please see response to earlier comment about the use of PREM.

**P7 Section 4.2 Coseismic Results Can you show the snapshots of the surface displacement?**

We have added 3 new figures to Appendix B showing coseismic surface displacement and postseismic surface displacement at the 3 times shown in profiles on figures 3-5. The 3 new figures correspond to the fault geometries tested.

**P7 Section 4.3 Postseismic Results Can you show the snapshots of the surface displacement at selected time steps?**

Please see response to previous comment.

**P7L214: "...less coseimic dispalcemetn from the ABAQUS. . ." What is the reason for less coseismic displacement for ABAQUS?**

We attribute this to mesh issues, as detailed on line 205 (new line 227). We have now included more details on mesh resolution tests as suggested by reviewer 2, which provides further justification. The text in Section 4.3 has been amended as follows:

*The mismatch in the coseismic displacement due to limitations in mesh resolution is the cause of mismatch for the postseismic displacement, i.e., less coseismic displacement from the ABAQUS model would result in less stress and therefore less relaxation. Slight improvements in the near-field displacement could be made by increasing the mesh resolution in the vicinity of the fault but would come at a computational cost (see Appendix A).*

**P9L251: "...an approximation of all the fault planes into a single geometry would be required.." I don't think this is a reliable way. A better alternative is to use the moment density tensor approach.**

Please refer to our earlier comment and manuscript edits regarding the moment-density tensor approach. We agree that in the near-field results would not be as reliable as fully representing the structure, however, representing complex fault geometry with a single plane geometry can provide a useful way of modelling far-field postseismic deformation. This method has been used by Takeuchi and Fialko (2013), and we will include a reference to this study on line 251 (new line 277).

**Figure 1: This figure may be sharper and better in black and white.**

When outputting graphics from ABAQUS in black and white the quality is worse. We will keep the colour but output at higher resolution.

**Figures 3-5: Showing the depth only to 100 km is confusing. Either show the full depth or explain it in the captions.**

We show the upper 100km as these material properties are those that primarily govern the Earth's response, but we agree this could be confusing. We have changed the caption to include the following sentence:

*Top panels of a) and b) show the fault dimensions and material properties of the upper 100 km of the model. Layers and material properties below 100 km depth are given in Table 2.*

**Figures 3-7: Figures look low in quality. It may be better to save those figures in vector graphics, if possible.**

Figures 3-7 will be submitted as vector graphics with the final manuscript.

**Finally, the following references are worth citing. References: Al-Attar, D. & Tromp, J., 2014: Sensitivity kernels for viscoelastic loading based on adjoint methods Geophysical Journal International, Oxford University Press, 196, 34-77**

**Dziewonski, A. M. & Anderson, D. L., 1981: Preliminary reference Earth model Physics of the Earth and Planetary Interiors, 25, 297-356**

**Latychev, K.; Mitrovica, J. X.; Tromp, J.; Tamisiea, M. E.; Komatitsch, D. & Christara, C. C., 2005: Glacial isostatic adjustment on 3-D Earth models: a finite-volume formulation Geophysical Journal International, Blackwell Science Ltd, 161, 421-444**

**Zhong, S.; Paulson, A. & Wahr, J., 2003: Three-dimensional finite-element modelling of Earth's viscoelastic deformation: effects of lateral variations in lithospheric thickness Geophysical Journal International, 155, 679-695**

We thank the reviewer for suggesting further references and will include them as appropriate in the text.

Additional References:

Gharti, H. N., Langer, L., and Tromp, J., 2019. Spectral-infinite-element simulations of earthquake-induced gravity perturbations, Geophysical Journal International, 217, 451-468. DOI:10.1093/gji/ggz028.

Khazaradze, G., Wang, K., Klotz, J., Hu, Y., and He, J., Prolonged post-seismic deformation of the 1960 great Chile earthquake and implications for mantle rheology, Geophys. Res. Lett., 29( 22), 2050, doi:10.1029/2002GL015986, 2002.

Tregoning, P., Burgette, R., McClusky, S. C., Lejeune, S., Watson, C. S., and McQueen, H. (2013), A decade of horizontal deformation from great earthquakes, J. Geophys. Res. Solid Earth, 118, 2371– 2381, doi:10.1002/jgrb.50154.

Zhou, X., Sun, W., Zhao, B., Fu, G., Dong, J., and Nie, Z. (2012), Geodetic observations detecting coseismic displacements and gravity changes caused by the Mw = 9.0 Tohoku-Oki earthquake, J. Geophys. Res., 117, B05408, doi:10.1029/2011JB008849.

**Response to Reviewer 2**

**Review of "A global, spherical, finite-element model for postseismic deformation using ABAQUS" by Nield and co-authors.**

We thank the reviewer for these suggestions and have responded to the detailed comments in line below.

**The manuscript represents an implementation of postseismic viscoelastic relaxation problems in a widely used finite-element commercial package. The study addresses common problems associated with meshing the domain, which is difficult around faults, and benchmarks the results against semi-analytic solutions attained with another widely used, but open-source, package. The study is accompanied with supplementary material that allows the community to reproduce and expand on these results quickly.**

**The study makes a number of simplifying assumptions about the rheology of the Earth that permits direct comparison with the semi-analytic code visco1d. However, once the code is benchmarked, these assumptions should be relaxed and more realistic constitutive laws that include a power-law stress/strain-rate relationship at steady state and a similar power-law constitutive behavior for transient creep - all compatible with laboratory observation of olivine creep - should be implemented and described. More realistic distributions of physical properties associated with thermal activation of viscoelastic flow in a realistic thermal field should follow.**

Power-law and transient power-law rheology have not been included in this study as the primary aim is to benchmark coseismic and postseismic displacement results against those produced by existing models with linear rheology. The implementation in ABAQUS of the rheologies mentioned by the reviewer is straight forward and has been done in other studies as mentioned on line 108. We have added more detail and references such as those suggested by the reviewer in their later comment to section 2.2 to expand on this (see extract below). However, using our model with more complex rheology will be the subject of future work and we feel that this is outside of the scope of our benchmarking study.

> *In this study we limit our benchmarking examples to a 1D linear viscoelastic rheology with one (Maxwell) or two (Burgers) relaxation times, however ABAQUS has the capability of implementing a variety of rheological models, including user-specified constitutive equations. For example, Freed et al. (2012) combined power-law rheology with a transient phase to model postseismic deformation following the 1999 magnitude 7.1 Hector Mine earthquake, and van der Wal et al. (2010) used a composite rheology based on laboratory-derived flow laws for diffusion and dislocation creep (Hirth and Kohlstedt, 2003; Karato and Wu, 1993) to model global glacial isostatic adjustment. Furthermore, variations of our model could be constructed using a 3D Earth structure (e.g. van der Wal et al., 2015).*

**A remaining issue is the meshing around more complex fault assembly that include multiple surfaces is not included in the model. As many earthquakes are now imaged to such a level of accuracy that these details are often well constrained, including complex fault geometry would be a relevant addition.**

We agree this remains a limitation of the model due to the difficulties in constructing a mesh around a complex fault structure with brick elements. However, we are focusing on the far-field postseismic displacement which is

less sensitive to simplifications made to the fault geometry than near-field displacement (Khazaradze et al., 2002, Tregoning et al., 2013, Zhou et al. 2012). Representing complex fault geometry with a single plane geometry can provide a useful way of modelling far-field postseismic deformation. This method has been used by Takeuchi and Fialko (2013), and we will include a reference to this study on line 251 (new line 277). When applied to case studies, fault and slip properties in the model can be adjusted so that model output matches observations of coseismic displacements which provides further confidence in modelled far-field deformations (e.g. Sun et al., 2018). We have amended the text in the discussion section to reflect these points:

> *In the case of a fault inversion that suggests multiple fault segments (e.g. Ye et al., 2014), an approximation of all the fault planes into a single geometry could still provide a realistic far-field estimate of postseismic deformation (e.g. Takeuchi and Fialko, 2013), particularly if the fault geometry and slip are adjusted so that model output matches observations of coseismic displacement (e.g. Sun et al., 2018). Far-field postseismic deformation is less sensitive to simplifications made to the fault geometry and slip distribution than near-field deformation (Khazaradze et al., 2002, Tregoning et al., 2013, Zhou et al. 2012).*

**Finally, the iterative procedure to include self-gravity should be replaced by directly solving the appropriate equations based on advection of pre-stress.**

This approach is not possible for a spherical model in ABAQUS, please also see more detailed response to comment below.

**I follow with a few detailed remarks.**

**55: An example of finite-element modeling of post-seismic relaxation with a spherical geometry is**

**Agata, R., Barbot, S.D., Fujita, K., Hyodo, M., Iinuma, T., Nakata, R., Ichimura, T. and Hori, T., 2019. Rapid mantle flow with power-law creep explains deformation after the 2011 Tohoku mega-quake.ÂaNature communications, ˘ a10(1), pp.1-11. ˘**

This additional reference will be included.

**105: Since it seems so easy to add more realistic rheology with the method, it should actually be done in this study. More realistic rheology involves a power-law stress/strain-rate relationship, see**

**Hirth, G. and Kohlstedt, D.L., 2003. Rheology of the Upper Mantle and the Mantle Wedge: A View from the Experimentalists: Inside the Subduction Factory, v. 138.**
**Karato, S.I. and Wu, P., 1993. Rheology of the upper mantle: A synthesis. Science, C2 GMDD Interactive comment Printer-friendly version Discussion paper 260(5109), pp.771-778.**
**Recent development include the inclusion of transient creep compatible with nonlinear steady-state creep:**
**Masuti, S., Barbot, S.D., Karato, S.I., Feng, L. and Banerjee, P., 2016. Upper-mantle water stratification inferred from observations of the 2012 Indian Ocean earthquake. Nature, 538(7625), pp.373-377.**
**Inclusion of realistic rheology seems more important and relevant than including selfgravitation.**

We feel this is outside the scope of the benchmarking study, please also refer to our earlier comment.

**130: It is unfortunate that Abaqus cannot simply solve the appropriate governing equations for self-gravitation and that these iterations are necessary. How can that be improved? Is there a way to solve a user-defined set of equations? Are the governing equations with self-gravitation not readily included in Abaqus? How is advection of pre-stress included?**

The governing equations solved in ABAQUS cannot be changed. A gravity load can be included directly within ABAQUS as a uniform acceleration in one fixed direction, therefore it is not easily applied to a spherical model. We choose instead to use the iterative approach as this has been shown by others (Wu, 2004) to correctly represent self-gravitation for a spherical viscoelastic Earth.

Advection of pre-stress is included via the elastic foundations described in section 2.3. We have added additional text and references to Wu (2004) to this section to clarify.

> *This means that advection of pre-stress is included and takes care of the restoring forces of buoyancy neglected in a conventional finite-element model (Wu, 2004, equation 3). The elastic foundations have a stiffness equal to the difference in density multiplied by gravitational acceleration (see Wu (2004) equations 12a,b,c).*

**180: The horizontal and vertical resolutions of the mesh seem inadequate to resolve the near field. The fault is 200x20 km and the mesh size around it is 10x5 km, representing just 20x4 mesh elements along the fault. It is actually surprising that the numerical result match the analytic solution so well with such a coarse mesh. This is perhaps an area of improvement.**

We are focusing on the far-field postseismic deformation within a global setting, which requires a mesh with a very large number of elements. It is therefore computationally expensive to have a very high-resolution mesh in the near-field and we are forced to trade off the mesh resolution (and hence more accurate near-field results) against computation time. To provide further justification for our choice of mesh resolution we have performed extra sensitivity tests for one of the fault cases and included the results as Appendix A. We have summarised the results in the text in Section 4.1 as follows:

> *For the 30° dipping reverse fault we tested a coarser and a finer mesh and found that our chosen mesh resolution provided a satisfactory trade-off between computation time and accuracy, with the finer mesh giving only small differences in near-field displacement compared to our chosen mesh, for a 10-fold increase in computation time. The results of this sensitivity test are discussed in more detail in Appendix A.*

**215: We need to see a convergence test in terms of mesh resolution for these cases. This may not necessitate more figures, but this needs to be discussed. I suspect that the resolution of the mesh in the near field can be improved, with valuable gains on the misfit. I suspect that the discrepancies that accumulate at long period during postseismic relaxation may be reduced with a more appropriate mesh in the near field.**

As per our response to the previous comment, we have performed extra sensitivity tests for mesh resolution and summarised the results in the text. A more detailed description of the test and results has been included as a new Appendix (A).

**250: If linear rheology models are assumed, several simulations can be run with separate parts of a complex fault geometry model - each fault at a time - and the results subsequently combined.**

This is an approach that could work for modelling complex faults with linear rheology, we have included the following sentence in the discussion section to include this point.

[revised manuscript text omitted]

---

## Referee Report (RR1)

The authors of this paper present a spherical self-gravitating finite element model generated using the commercial code ABAQUS. The main purpose of the paper is to provide a numerical platform (ABAQUS input files) and associated benchmark useful for future studies of far-field postseismic (viscoelastic) deformation triggered by large earthquakes. A Spherical domain is required to overcome the limits imposed by flat surface analytical or semi-analytical solutions, and self-gravitation is important especially when we want to combine the deformation field from sources other than earthquakes (e.g. post-glacial rebound) or for multi-cycle simulations.

The ABAQUS input file presented in the paper is structured in two parts. The first part (*Static) controls the purely elastic coseismic phase and the implementation of fault slip using kinematic constraint equations. The second part (*Visco) controls the long term postseismic phase (viscoelastic relaxation) using Maxwell or Burgers rheology. The structure of the ABAQUS file is user friendly because provides a clear separation between the different phases of the simulation and is generally easy to modify from other people.

Large part of the paper is correctly focused on the benchmark of the coseismic and postseismic simulation against known analytical solutions. More specifically, for the coseismic phase, the authors compare the ABAQUS predicted displacement field with the well known Okada (1985) solutions for a rectangular source implemented in a homogeneous isotropic half-space. They show a general good agreement between the numerical and the analytical approaches. For the postseismic viscoelastic deformation they compare the ABAQUS predictions with the spherical harmonics code VISCO1D, showing again good agreement. The correct reproduction of analytical solutions using ABAQUS is very useful and proves the correct implementation of fault slip and viscoelastic relaxation.

In my opinion, this paper (technical activity) requires one or two additions before publication. I explain my point below.

I commend the authors for working on a such important and complex topic. However, I believe there is at least one main issue to be discussed before publication.

1. I think it will be helpful for future users of your ABAQUS input files if you could provide at least one forward model that implements a different rheologic structure. For example PREM or any recent model with lateral variations in material properties (e.g. viscosity). This way any future user will have an example to learn and to modify for its own project. The same thing is noted from Reviewer 1 although you have not updated your work to accommodate this specific request. You have already successfully completed the benchmark. For that reason, you don't need to benchmark again against a code that implements PREM (although it will be useful). I feel that this work will be significantly improved if you present what described above.

2. I'm not sure If I completely understand the problem regarding the implementation of multiple faults. If I understand correctly, implementing multiple faults will require a verification of the mesh quality. Verifying the quality of the mesh is part of the FE modeling work anyway. ABAQUS will give you a warning if the elements are distorted, although it will most probably finish the calculation. You can easily implement multiple sources (or variable fault geometry) in ABAQUS, you will just have to take the time to verify your mesh using the appropriate metrics. Another solution will be to use an external meshing software like Trelis/Cubit, that will give you better control on the generation of the mesh, and subsequently use this mesh to generate the ABAQUS input file. I will agree however, that for the purpose of investigating far-field deformation, details regarding the fault geometry might be of secondary order.

---

## Author Response (AR2)

**Response to Editor**

Grace

The reviewers' responses to your revised manuscript are available now.

Could you please address their commends?

I would like to emphasize that both reviewers (as also suggested by a reviewer in round 1) have recommended to add a case including a PREM based material model. This would significantly increase the relevance of the paper as PREM is used as a standard model in the literature. Also as highlighted by reviewer 2 providing access to the data file would be tremendously helpful for readers considering using Abaqus which would further increase the relevance of the work.

Thanks.

Lutz Gross

GMD Topical Editor
* * *
Dear Lutz,

Thank you for the reviews for our revised manuscript, we are grateful for further constructive comments. We have taken onboard the comment made by both reviewers regarding a PREM model and have implemented this earth model as an additional Abaqus test case. The results are included in Figs. 6a, 7a and B4 and discussed in the text (detailed responses are provided below). The Abaqus input file for this run has been added to the online repository, and furthermore we have added information to the Abaqus README text file to describe how future users may implement their own laterally varying earth models.

Please find in this final response a point-by-point reply (plain black text) to all reviewer comments (red text), including the changes we have made to the manuscript (marked in italics). The revised manuscript is attached below the response to the reviewers.

Please also note we have changed reference to Abaqus from upper case to lower case under advisement and have italicised "Abaqus" and "VISCO1D" in the text.

Kind Regards

Grace Nield & co-authors

**Response to Reviewer 1**

I would like to thank and acknowledge the effort made by authors in revising their manuscript. They have addressed most of my concerns, and I am mostly satisfied with their response although there are some misleading statements.

We thank the reviewer for taking the time to re-review our manuscript, and for the positive response to our amendments. We have responded to each comment in line below.

"We have not included the governing equations for the finite element formulation as this part of the study is not new work. The model is based on the finite element formulation of Wu (2004) and equations therein, as referenced on line 84. Instead, we focus the paper on describing the new aspects of this model –incorporating a fault plane and prescribing slip."

The article Agata et al. 2019 cited by the authors seems to be implementing a fault plane and prescribing slip, not only for forward modelling but also for adjoint simulations.

Then readers will probably ask: what is the new contribution in this article as compared to Wu 2004 and Agata 2019? In this context, readers would appreciate some clarity on this.

The main difference between our model and that of Agata et al. (2019) is that we present a fully global self-gravitational model and Agata et al. (2019) restrict the spatial extent of their model to a local domain. We have clarified on line 57 that their model is not global.

Despite this question, I recognize the importance of this work, and therefore, I am willing to recommend this article for publication after a minor revision.

Minor comments:

"Using the Preliminary Reference Earth Model (PREM) instead would not add anything to the results in our opinion, and only complicate reproducibility of the results for others."

I disagree, in particular with "...only complicate reproducibility of the results for others." In fact, the PREM is the standard model for many geophysical simulations on a global scale.

We have taken on board this comment and have implemented a new Abaqus model using the PREM elastic properties along with viscosity profile from the existing benchmarking tests. We have not benchmarked this test case as the Okada analytical solution is valid only for a homogenous earth and would not provide a valid comparison. The results from this new model run are included in Figure 6a and Figure 7a, which now show displacement through time for the fault geometry in test case 3 with 3 different earth models: The simple earth model with Maxwell and Burgers rheology, and the PREM model. Furthermore, the surface displacement from this new model is also included in Appendix B (Fig. B4). The Abaqus input file has been added to the supplementary information.

We have added the following text to Section 4.1:

> "Furthermore, we run the third test case in Abaqus using the elastic properties from PREM, keeping the same three-layer Maxwell viscosity profile as the benchmarking Earth model (Table 2). We do not benchmark the results from this test case as the Okada analytical solution is only valid for a homogeneous Earth."

The following paragraph has been added to Section 4.3:

> "Figs. 6a and 7a also show the displacement for test case 3 with the PREM elastic properties (green dotted lines). At 100 km distance from the fault (Fig, 6a), the different elastic structure results in slightly more coseismic displacement than the simple elastic structure used in the benchmarking tests, which results in greater postseismic displacement with time. At 300 km distance from the fault (Fig. 7a), the depth varying PREM elastic structure results in only very small differences. The surface coseismic and postseismic displacements at 10, 50 and 100 years after the earthquake are shown in Fig. B4, Appendix B."

**"We will change this term to "layered halfspace" but will also note on first use that it is also referred to as "flat-Earth" in other literature (e.g. Wu,2004) so as not to cause confusion to the reader."**

**Personally, I am not against using the term "flat-Earth", but these days, it seems to be associated more with the flat-earth conspiracies:)**

Yes, we agree, and the term "layered half-space" is a good alternative.

**"The moment-density tensor approach suggested by the reviewer is an alternative method of representing a fault within a spectral-element mesh (e.g., Gharti et al. 2019) whereby the mesh geometry is not required to conform to the geometry of the fault plane. However, our current knowledge is that this method cannot be implemented in a finite-element mesh in ABAQUS due to the restrictions of defining loads and forces on elements or surfaces within the mesh."**

**I think this is untrue. Because the moment-density tensor approach is well adapted to weak formulations and ABAQUS being a finite-element tool, implementing the moment-density tensor approach in ABAQUS should be straightforward.**

We concede there is a gap in our knowledge regarding the moment-density tensor approach. From research that we have done in response to this comment, we cannot ascertain how to implement this approach in Abaqus. It may be possible, but it is not clear in the documentation or from other literature, and therefore we feel reluctant to include this as an option within the manuscript.

**Table A1.**

**Are those computation times for single or parallel processing?**

The times quoted in Table A1 are for parallel processing on the 8 cores of the machine. We have added this into the text on new line 398.

**Response to Reviewer 2**

The authors of this paper present a spherical self-gravitating finite element model generated using the commercial code ABAQUS. The main purpose of the paper is to provide a numerical platform (ABAQUS input files) and associated benchmark useful for future studies of far-field postseismic (viscoelastic) deformation triggered by large earthquakes. A Spherical domain is required to overcome the limits imposed by flat surface analytical or semi-analytical solutions, and self-gravitation is important especially when we want to combine the deformation field from sources other than earthquakes (e.g. post-glacial rebound) or for multi-cycle simulations.

The ABAQUS input file presented in the paper is structured in two parts. The first part (*Static) controls the purely elastic coseismic phase and the implementation of fault slip using kinematic constraint equations. The second part (*Visco) controls the long term postseismic phase (viscoelastic relaxation) using Maxwell or Burgers rheology. The structure of the ABAQUS file is user friendly because provides a clear separation between the different phases of the simulation and is generally easy to modify from other people.

Large part of the paper is correctly focused on the benchmark of the coseismic and postseismic simulation against known analytical solutions. More specifically, for the coseismic phase, the authors compare the ABAQUS predicted displacement field with the well known Okada (1985) solutions for a rectangular source implemented in a homogeneous isotropic half-space. They show a general good agreement between the numerical and the analytical approaches. For the postseismic viscoelastic deformation they compare the ABAQUS predictions with the spherical harmonics code VISCO1D, showing again good agreement. The correct reproduction of analytical solutions using ABAQUS is very useful and proves the correct implementation of fault slip and viscoelastic relaxation.

In my opinion, this paper (technical activity) requires one or two additions before publication. I explain my point below.

I commend the authors for working on a such important and complex topic. However, I believe there is at least one (point 1 below) main issue to be discussed and solved before publication.

We thank the reviewer for providing helpful comments and suggestions for the manuscript and we have responded in line below.

1. I think it will more helpful for future users of your examples "ABAQUS input files" if you could provide at least one forward model that implements a different rheologic structure. For example PREM or any recent model with lateral variations in material properties (e.g. viscosity). This way any future user will have an example to learn and to modify for its own project. The same thing is noted from Reviewer 1 although you have not updated your work to accommodate this specific request. You have already successfully completed the benchmark. For that reason, you don't need to benchmark again against a code that implements PREM (although it will be useful). I feel that this work will be significantly improved if you present what described above.

We acknowledge this comment from both reviewers and have included an additional Abaqus run for the fault geometry in test case 3, using the PREM elastic structure along with the viscosity profile from the existing earth model presented in the paper. We have chosen not to benchmark this case as the Okada analytical solution is valid only for a homogenous earth, and VISCO1D does not lend itself easily to the number of layers needed to model PREM.

The results have been added to Figure 6a and Figure 7a and we have included the surface displacement in Appendix B (Fig. B4).

We have added the following text to Section 4.1:

> "Furthermore, we run the third test case in Abaqus using the elastic properties from PREM, keeping the same three-layer Maxwell viscosity profile as the benchmarking Earth model (Table 2). We do not benchmark the results from this test case as the Okada analytical solution is only valid for a homogeneous Earth."

The following paragraph has been added to Section 4.3:

> "Figs. 6a and 7a also show the displacement for test case 3 with the PREM elastic properties (green dotted lines). At 100 km distance from the fault (Fig, 6a), the different elastic structure results in slightly more coseismic displacement than the simple elastic structure used in the benchmarking tests, which results in greater postseismic displacement with time. At 300 km distance from the fault (Fig. 7a), the depth varying PREM elastic structure results in only very small differences. The surface coseismic and postseismic displacements at 10, 50 and 100 years after the earthquake are shown in Fig. B4, Appendix B."

The input file for this extra model run has been added to the online repository. Furthermore, to aid future users, we have included extra information in the Abaqus README file in this repository to describe how future users would implement different earth structures such as one containing lateral variations in viscosity.

**2. I'm not sure If I completely understand the problem regrading the implementation of multiple faults. If I understand correctly, implementing multiple faults will require a verification of the mesh quality. Verifying the quality of the mesh is part of the FE modeling work anyway. ABAQUS will give you a warning if the elements are distorted, although it will most probably finish the calculation. You can easily implement multiple sources (or variable fault geometry) in ABAQUS, you will just have to take the time to verify your mesh using the appropriate metrics. Another solution will be to use an external meshing software like Trelis/Cubit, that will give you better control on the generation of the mesh, and subsequently use this mesh to generate the ABAQUS input file. I will agree however, that for the purpose of investigating far-field deformation, details regarding the fault geometry might be of secondary order.**

Yes, this understanding is correct. One of the difficulties in Abaqus is to construct the geometry of a layered spherical Earth with more than one fault plane transecting the uppermost layer because this needs to be done with the accuracy of a specific strike/dip/length/width of fault plane. In addition to this, many partitions are needed to enable the software to mesh the given geometry. This is not a trivial problem. Our aim was to make a code for Abaqus that would automatically generate the model for a fault plane with different strike, dip, length, and width and at present this does not work with 2 or more fault planes.

We agree with the reviewer, however, that it could be done by re-writing the code, or by implementing external meshing software. We have rephrased the language used in the manuscript (line 284-288) to reflect this as follows:

[revised manuscript text omitted]